# Subspace Recovery from Heterogeneous Data with Non-isotropic Noise

**John Duchi**
Stanford University
jduchi@stanford.edu

**Vitaly Feldman**
Apple
vitaly.edu@gmail.com

**Lunjia Hu**
Stanford University
lunjia@stanford.edu

**Kunal Talwar**
Apple
kunal@kunaltalwar.org

## Abstract

Recovering linear subspaces from data is a fundamental and important task in statistics and machine learning. Motivated by heterogeneity in Federated Learning settings, we study a basic formulation of this problem: the principal component analysis (PCA), with a focus on dealing with irregular noise. Our data come from $n$ users with user $i$ contributing data samples from a $d$-dimensional distribution with mean $\mu_i$. Our goal is to recover the linear subspace shared by $\mu_1, \ldots, \mu_n$ using the data points from all users, where every data point from user $i$ is formed by adding an independent mean-zero noise vector to $\mu_i$. If we only have one data point from every user, subspace recovery is information-theoretically impossible when the covariance matrices of the noise vectors can be non-spherical, necessitating additional restrictive assumptions in previous work. We avoid these assumptions by leveraging at least two data points from each user, which allows us to design an efficiently-computable estimator under non-spherical and user-dependent noise. We prove an upper bound for the estimation error of our estimator in general scenarios where the number of data points and amount of noise can vary across users, and prove an information-theoretic error lower bound that not only matches the upper bound up to a constant factor, but also holds even for spherical Gaussian noise. This implies that our estimator does not introduce additional estimation error (up to a constant factor) due to irregularity in the noise. We show additional results for a linear regression problem in a similar setup.

## 1 Introduction

We study the problem of learning low-dimensional structure amongst data distributions, given multiple samples from each distribution. This problem arises naturally in settings such as federated learning, where we want to learn from data coming from a set of individuals, each of which has samples from their own distribution. These distributions however are related to each other, and in this work, we consider the setting when these distributions have means lying in a low-dimensional subspace. The goal is to learn this subspace, even when the distributions may have different (and potentially non-spherical) variances. This heterogeneity can manifest itself in practice as differing number of samples per user, or the variance differing across individuals, possibly depending on their mean. Recovery of the subspace containing the means can in turn help better estimate individual means. In other words, this can allow for learning good estimator for all individual means, by leveraging information from all the individuals.

36th Conference on Neural Information Processing Systems (NeurIPS 2022).

The irregularity of the noise makes this task challenging even when we have sufficiently many individual distributions. For example, suppose we have $n$ individuals and for every $i = 1, \ldots, n$, an unknown $\mu_i \in \mathbb{R}^d$. For simplicity, suppose that $\mu_1, \ldots, \mu_n$ are distributed independently as $N(0, \sigma^2 u u^\mathsf{T})$ for $\sigma \in \mathbb{R}_{\geq 0}$ and an unknown unit vector $u \in \mathbb{R}^d$. In this setting, our goal is to recover the one-dimensional subspace, equivalently the vector $u$. For every $i$, we have a data point $x_i = \mu_i + z_i$ where $z_i \in \mathbb{R}^d$ is a mean-zero noise vector. If $z_i$ is drawn independently from a spherical Gaussian $N(0, \alpha^2 I)$, we can recover the unknown subspace with arbitrary accuracy as $n$ grows to infinity because $\frac{1}{n} \sum x_i x_i^\mathsf{T}$ concentrates to $\mathbb{E}[x_i x_i^\mathsf{T}] = \sigma^2 u u^\mathsf{T} + \alpha^2 I$, whose top eigenvector is $\pm u$. However, if the noise $z_i$ is drawn from a non-spherical distribution, the top eigenvector of $\frac{1}{n} \sum x_i x_i^\mathsf{T}$ can deviate from $\pm u$ significantly, and to make things worse, if the noise $z_i$ is drawn independently from a non-spherical Gaussian $N(0, \sigma^2(I - u u^\mathsf{T}) + \alpha^2 I)$, then our data points $x_i = \mu_i + z_i$ distribute independently as $N(0, (\sigma^2 + \alpha^2)I)$, giving no information about the vector $u$.[1]

The information-theoretic impossibility in this example however disappears as soon as one has at least two samples from each distribution. Indeed, given two data points $x_{i1} = \mu_i + z_{i1}$ and $x_{i2} = \mu_i + z_{i2}$ from user $i$, as long as the noise $z_{i1}, z_{i2}$ are independent and have zero mean, we always have $\mathbb{E}[x_{i1} x_{i2}^\mathsf{T}] = \sigma^2 u u^\mathsf{T}$ regardless of the specific distributions of $z_{i1}$ and $z_{i2}$. This allows us to recover the subspace in this example, as long as we have sufficiently many users each contributing at least two examples.

As this is commonly the case in our motivating examples, we make this assumption of multiple data points per user, and show that this intuition extends well beyond this particular example. We design efficiently computable estimators for this subspace recovery problem given samples from multiple heteroscedastic distributions (see Section 1.1 for details). We prove upper bounds on the error of our estimator measured in the *maximum principal angle* (see Section 2 for definition). We also prove an information-theoretic error lower bound, showing that our estimator achieves the optimal error up to a constant factor in general scenarios where the number of data points and the amount of noise can vary across users. Somewhat surprisingly, our lower bound holds even when the noise distributes as spherical Gaussians. Thus non-spherical noise in setting does not lead to increased error.

We then show that our techniques extend beyond the mean estimation problem to a linear regression setting where for each $\mu_i$, we get (at least two) samples $(x_{ij}, x_{ij}^\mathsf{T} \mu_i + z_{ij})$ where $z_{ij}$ is zero-mean noise from some noise distribution that depends on $i$ and $x_{ij}$. This turns out to be a model that was recently studied in the meta-learning literature under more restrictive assumptions (e.g. $z_{ij}$ is independent of $x_{ij}$) [Kong et al., 2020, Tripuraneni et al., 2021, Collins et al., 2021, Thekumparampil et al., 2021]. We show a simple estimator achieving an error upper bound matching the ones in prior work without making these restrictive assumptions.

## 1.1 Our contributions

**PCA with heterogeneous and non-isotropic noise: Upper Bounds.** In the PCA setting, the data points from each user $i$ are drawn from a user-specific distribution with mean $\mu_i \in \mathbb{R}^d$, and we assume that $\mu_1, \ldots, \mu_n$ lie in a shared $k$-dimensional subspace that we want to recover. Specifically, we have $m_i$ data points $x_{ij} \in \mathbb{R}^d$ from user $i$ for $j = 1, \ldots, m_i$, and each data point is determined by $x_{ij} = \mu_i + z_{ij}$ where $z_{ij} \in \mathbb{R}^d$ is a noise vector drawn independently from a mean zero distribution. We allow the distribution of $z_{ij}$ to be non-spherical and non-identical across different pairs $(i, j)$. We use $\eta_i \in \mathbb{R}_{\geq 0}$ to quantify the amount of noise in user $i$'s data points by assuming that $z_{ij}$ is an $\eta_i$-*sub-Gaussian* random variable.

As mentioned earlier, if we only have a single data point from each user, it is information-theoretically impossible to recover the subspace. Thus, we focus on the case where $m_i \geq 2$ for every $i = 1, \ldots, n$. In this setting, for appropriate weights $w_1, \ldots, w_n \in \mathbb{R}_{\geq 0}$, we compute a matrix $A$:

$$A = \sum_{i=1}^n \frac{w_i}{m_i(m_i - 1)} \sum_{j_1 \neq j_2} x_{ij_1} x_{ij_2}^\mathsf{T}, \tag{1}$$

where the inner summation is over all pairs $j_1, j_2 \in \{1, \ldots, m_i\}$ satisfying $j_1 \neq j_2$. Our estimator is then defined by the subspace spanned by the top-$k$ eigenvectors of $A$. Although the inner summation

---

[1]This information-theoretic impossibility naturally extends to recovering $k$-dimensional subspaces for $k > 1$ by replacing the unit vector $u \in \mathbb{R}^d$ with a matrix $U \in \mathbb{R}^{d \times k}$ with orthonormal columns.

is over $m_i(m_i - 1)$ terms, the time complexity for computing it need not grow quadratically with $m_i$ because of the following equation:

$$\sum_{j_1 \neq j_2} x_{ij_1} x_{ij_2}^{\mathsf{T}} = \left(\sum_{j=1}^{m_i} x_{ij}\right)\left(\sum_{j=1}^{m_i} x_{ij}\right)^{\mathsf{T}} - \sum_{j=1}^{m_i} x_{ij} x_{ij}^{\mathsf{T}}.$$

The flexibility in the weights $w_1, \ldots, w_n$ allows us to deal with variations in $m_i$ and $\eta_i$ for different users $i$. In the special case where $\eta_1 = \cdots = \eta_n = \eta$ and $m_1 = \cdots = m_n = m$, we choose $w_1 = \cdots = w_n = 1/n$ and we show that our estimator achieves the following error upper bound with success probability at least $1 - \delta$:

$$\sin \theta = O\left(\left(\frac{\eta \sigma_1}{\sigma_k^2 \sqrt{m}} + \frac{\eta^2}{\sigma_k^2 m}\right)\sqrt{\frac{d + \log(1/\delta)}{n}}\right).$$

Here, $\theta$ is the maximum principal angle between our estimator and the true subspace shared by $\mu_1, \ldots, \mu_n$, and we define $\sigma_\ell \geq 0$ such that $\sigma_\ell^2$ is the $\ell$-th largest eigenvalue of $\sum_{i=1}^n w_i \mu_i \mu_i^{\mathsf{T}}$. Our error upper bound for general $m_i, \eta_i, w_i$ is given in Theorem 3.1.

We instantiate our error upper bound to the case where $\mu_1, \ldots, \mu_n$ are drawn iid from a Gaussian distribution $N(0, \sigma^2 U U^{\mathsf{T}})$, where the columns of $U \in \mathbb{R}^{d \times k}$ form an orthonormal basis of the subspace containing $\mu_1, \ldots, \mu_n$. By choosing the weights $w_1, \ldots, w_n$ according to $m_1, \ldots, m_n$ and $\eta_1, \ldots, \eta_n$, our estimator achieves the error upper bound

$$\sin \theta \leq O\left(\sqrt{\frac{d + \log(1/\delta)}{\sum_{i=1}^n \gamma_i'}}\right) \tag{2}$$

under a mild assumption (Assumption 3.2), where $\gamma_i'$ is defined in Definition 3.1 and often equals $\left(\frac{\eta_i^2}{\sigma^2 m_i} + \frac{\eta_i^4}{\sigma^4 m_i^2}\right)^{-1}$.

**PCA: Lower Bounds.** We show that the error upper bound (2) is optimal up to a constant factor by proving a matching information-theoretic lower bound (Theorem 3.7). Our lower bound holds for general $m_i$ and $\eta_i$ that can vary among users $i$, and it holds even when the noise vectors $z_{ij}$ are drawn from spherical Gaussians, showing that our estimator essentially pays no additional cost in error or sample complexity due to non-isotropic noise.

We prove the lower bound using Fano's method on a local packing over the Grassmannian manifold. We carefully select a non-trivial hard distribution so that the strength of our lower bound is not affected by a group of fewer than $k$ users each having a huge amount of data points with little noise.

**Linear Models.** While the PCA setting is the main focus of our paper, we extend our research to a related linear models setting that has recently been well studied in the meta-learning and federated learning literature [Kong et al., 2020, Tripuraneni et al., 2021, Collins et al., 2021, Thekumparampil et al., 2021]. Here, the user-specific distribution of each user $i$ is parameterized by $\beta_i \in \mathbb{R}^d$, and we again assume that $\beta_1, \ldots, \beta_n$ lie in a $k$-dimensional linear subspace that we want to recover. From each user $i$ we observe $m_i$ data points $(x_{ij}, y_{ij}) \in \mathbb{R}^d \times \mathbb{R}$ for $j = 1, \ldots, m_i$ drawn from the user-specific distribution satisfying $y_{ij} = x_{ij}^{\mathsf{T}} \beta_i + z_{ij}$ for an $O(1)$-sub-Gaussian measurement vector $x_{ij} \in \mathbb{R}^d$ with zero mean and identity covariance and an $\eta_i$-sub-Gaussian mean-zero noise term $z_{ij} \in \mathbb{R}$. While it may seem that non-isotropic noise is less of a challenge in this setting since each noise term $z_{ij}$ is a scalar, our goal is to handle a challenging scenario where the variances of the noise terms $z_{ij}$ can depend on the *realized* measurements $x_{ij}$, which is a more general and widely applicable setting compared to those in prior work. Similarly to the PCA setting, our relaxed assumptions on the noise make it information-theoretically impossible to do subspace recovery if we only have one data point from each user (see Section 4), and thus we assume each user contributes at least two data points. For appropriate weights $w_1, \ldots, w_n \in \mathbb{R}_{\geq 0}$, we use the subspace spanned by the top-$k$ eigenvectors of the following matrix $A$ as our estimator:

$$A = \sum_{i=1}^n \frac{w_i}{m_i(m_i - 1)} \sum_{j_1 \neq j_2} (x_{ij_1} y_{ij_1})(x_{ij_2} y_{ij_2})^{\mathsf{T}}. \tag{3}$$

In the special case where $\eta_1 = \cdots = \eta_n = \eta, m_1 = \cdots = m_n = m$, and $\|\beta_i\|_2 \leq r$ for all $i$, our estimator achieves the following error upper bound using weights $w_1 = \cdots = w_n = 1/n$:

$$\sin\theta \leq O\left(\log^3(nd/\delta)\sqrt{\frac{d(r^4 + r^2\eta^2 + \eta^4/m)}{mn\sigma_k^4}}\right), \tag{4}$$

where $\theta$ is the maximum principal angle between our estimator and the true subspace shared by $\beta_1, \ldots, \beta_n$, and $\sigma_k^2$ is the $k$-th largest eigenvalue of $\sum_{i=1}^n w_i\beta_i\beta_i^{\mathsf{T}}$ (Corollary L.2). Our error upper bound extends smoothly to more general cases where $\eta_i$ and $m_i$ vary among users (Theorem L.1). Moreover, our upper bound matches the ones in prior work [e.g. Tripuraneni et al., 2021, Theorem 3] despite requiring less restrictive assumptions.

## 1.2 Related Work

Principal component analysis under non-isotropic noise has been studied by Vaswani and Narayana-murthy [2017], Zhang et al. [2018] and Narayanamurthy and Vaswani [2020]. When translated to our setting, these papers focus on having only one data point from each user and thus they require additional assumptions—either the level of non-isotropy is low, or the noise is coordinate-wise independent and the subspace is incoherent. The estimation error guarantees in these papers depend crucially on how well these additional assumptions are satisfied. Zhu et al. [2019] and Cai et al. [2021] study PCA with noise and missing data, and Chen et al. [2021] and Cheng et al. [2021] study eigenvalue and eigenvector estimation under heteroscedastic noise. These four papers all assume that the noise is coordinate-wise independent and the subspace/eigenspace is incoherent.

The linear models setting we consider has recently been studied as a basic setting of meta-learning and federated learning by Kong et al. [2020], Tripuraneni et al. [2021], Collins et al. [2021], and Thekumparampil et al. [2021]. These papers all make the assumption that the noise terms $z_{ij}$ are independent of the measurements $x_{ij}$, an assumption that we relax in this paper. Collins et al. [2021] and Thekumparampil et al. [2021] make improvements in sample complexity and error guarantees compared to earlier work by Kong et al. [2020] and Tripuraneni et al. [2021], but Collins et al. [2021] focus on the noiseless setting ($z_{ij} = 0$) and Thekumparampil et al. [2021] require at least $\Omega(k^2)$ examples per user. Tripuraneni et al. [2021] and Thekumparampil et al. [2021] assume that the measurements $x_{ij}$ are drawn from the standard (multivariate) Gaussian distribution, where as Kong et al. [2020], Collins et al. [2021] and our work make the relaxed assumption that $x_{ij}$ are sub-Gaussian with identity covariance, which, in particular, allows the fourth-order moments of $x_{ij}$ to be non-isotropic. There is a large body of prior work on meta-learning beyond the linear setting [see e.g. Maurer et al., 2016, Tripuraneni et al., 2020, Du et al., 2020].

When collecting data from users, it is often important to ensure that private information about users is not revealed through the release of the learned estimator. Many recent works proposed and analyzed estimators that achieve *user-level differential privacy* in settings including mean estimation [Levy et al., 2021, Esfandiari et al., 2021], meta-learning [Jain et al., 2021] and PAC learning [Ghazi et al., 2021]. Recently, Cummings et al. [2021] study one-dimensional mean estimation in a setting similar to ours, under a differential privacy constraint.

The matrix $A$ we define in (1) is a weighted sum of $A_i := \frac{1}{m_i(m_i-1)}\sum_{j_1 \neq j_2} x_{ij_1}x_{ij_2}^{\mathsf{T}}$ over users $i = 1, \ldots, n$, and each $A_i$ has the form of a $U$-statistic [Halmos, 1946, Hoeffding, 1948]. $U$-statistics have been applied to many statistical tasks including tensor completion [Xia and Yuan, 2019] and various testing problems [Zhong and Chen, 2011, He et al., 2021, Schrab et al., 2022]. In our definition of $A_i$, we do *not* make the assumption that the distributions of $x_{i1}, \ldots, x_{im_i}$ are identical although the assumption is commonly used in applications of $U$-statistics. The matrix $A$ in (3) is also a weighted sum of $U$-statistics where we again do not make the assumption of identical distribution.

## 1.3 Paper Organization

In Section 2, we formally define the maximum principal angle and other notions we use throughout the paper. Our results in the PCA setting and the linear models setting are presented in Sections 3 and 4, respectively. We defer most technical proofs to the appendices.

## 2 Preliminaries

We use $\|A\|$ to denote the spectral norm of a matrix $A$, and use $\|u\|_2$ to denote the $\ell_2$ norm of a vector $u$. For positive integers $k \leq d$, we use $\mathbb{O}_{d,k}$ to denote the set of matrices $A \in \mathbb{R}^{d \times k}$ satisfying $A^\mathsf{T} A = I_k$, where $I_k$ is the $k \times k$ identity matrix. We use $\mathbb{O}_d$ to denote $\mathbb{O}_{d,d}$, which is the set of $d \times d$ orthogonal matrices. We use $\mathrm{col}(A)$ to denote the linear subspace spanned by the columns of a matrix $A$. We use the base-$e$ logarithm throughout the paper.

**Maximum Principal Angle.**   Let $U, \hat{U} \in \mathbb{O}_d$ be two orthogonal matrices. Suppose the columns of $U$ and $\hat{U}$ are partitioned as $U = [U_1\ U_2], \hat{U} = [\hat{U}_1\ \hat{U}_2]$ where $U_1, \hat{U}_1 \in \mathbb{O}_{d,k}$ for an integer $k$ satisfying $0 < k < d$. Let $\Gamma$ (resp. $\hat{\Gamma}$) be the $k$-dimensional linear subspace spanned by the columns of $U_1$ (resp. $\hat{U}_1$). Originating from [Jordan, 1875], the *maximum principal angle* $\theta \in [0, \pi/2]$ between $\Gamma$ and $\hat{\Gamma}$, denoted by $\angle(\Gamma, \hat{\Gamma})$ or $\angle(U_1, \hat{U}_1)$, is defined by $\sin\theta = \|U_1 U_1^\mathsf{T} - \hat{U}_1 \hat{U}_1^\mathsf{T}\| = \|U_1^\mathsf{T} \hat{U}_2\| = \|U_2^\mathsf{T} \hat{U}_1\|$. It is not hard to see that the maximum principal angle depend only on the subspaces $\Gamma, \hat{\Gamma}$ and not on the choices of $U$ and $\hat{U}$, and $\sin\angle(\Gamma, \hat{\Gamma})$ is a natural metric between $k$-dimensional subspaces (see Appendix A for more details where we discuss the definition of principal angles for any two subspaces with possibly different dimensions).

With the definition of the maximum principal angle, we can now state a variant of the Davis–Kahan $\sin\theta$ theorem [Davis and Kahan, 1970] that will be useful in our analysis (see Appendix E for proof):

**Theorem 2.1** (Variant of Davis–Kahan $\sin\theta$ theorem). *Let $A, \hat{A} \in \mathbb{R}^{d \times d}$ be symmetric matrices. Let $\lambda_i$ denote the $i$-th largest eigenvalue of $A$. For a positive integer $k$ smaller than $d$, let $\theta$ denote the maximum principal angle between the subspaces spanned by the top-$k$ eigenvectors of $A$ and $\hat{A}$. Assuming $\lambda_k > \lambda_{k+1}$,*

$$\sin\theta \leq \frac{2\|A - \hat{A}\|}{\lambda_k - \lambda_{k+1}}.$$

**Sub-Gaussian and sub-exponential distributions.**   We say a random variable $x \in \mathbb{R}$ with expectation $\mathbb{E}[x] \in \mathbb{R}$ has *sub-Gaussian* constant $b \in \mathbb{R}_{\geq 0}$ if $\mathbb{E}[|x - \mathbb{E}[x]|^p]^{1/p} \leq b\sqrt{p}$ for every $p \geq 1$. We say $x$ has *sub-exponential* constant $b \in \mathbb{R}_{\geq 0}$ if $\mathbb{E}[|x - \mathbb{E}[x]|^p]^{1/p} \leq bp$ for every $p \geq 1$. We say a random vector $y \in \mathbb{R}^d$ has sub-Gaussian (resp. sub-exponential) constant $b \in \mathbb{R}_{\geq 0}$ if for every unit vector $u \in \mathbb{R}^d$ (i.e., $\|u\|_2 = 1$), the random variable $u^\mathsf{T} y \in \mathbb{R}$ has sub-Gaussian (resp. sub-exponential) constant $b$. We say $y$ is $b$-*sub-Gaussian* (resp. $b$-*sub-exponential*) if it has sub-Gaussian (resp. sub-exponential) constant $b$.

## 3 Principal Component Analysis

In the principal component analysis (PCA) setting, our goal is to recover the $k$-dimensional subspace $\Gamma$ spanned by the user-specific means $\mu_1, \ldots, \mu_n \in \mathbb{R}^d$ of the $n$ users. From each user $i$, we have $m_i \geq 2$ data points

$$x_{ij} = \mu_i + z_{ij} \quad \text{for } j = 1, \ldots, m_i. \tag{5}$$

We assume the noise $z_{ij} \in \mathbb{R}^d$ is drawn independently from a mean zero distribution with sub-Gaussian constant $\eta_i$. We do *not* assume that the variance of $z_{ij}$ is the same along every direction, *nor* do we assume that the distribution of $z_{ij}$ is the same for different $(i, j)$. We first show an error upper bound for our estimator when the user-specific means $\mu_1, \ldots, \mu_n$ are deterministic vectors (Section 3.1) and then apply this result to the case where $\mu_1, \ldots, \mu_n$ are drawn from a sub-Gaussian distribution (Section 3.2). In Section 3.3 we prove an information-theoretic error lower bound matching our upper bound.

### 3.1 Fixed User-Specific Means

We first focus on the case where $\mu_1, \ldots, \mu_n$ are deterministic vectors. In this case, all the randomness in the data comes from the noise $z_{ij}$. Our estimator is the subspace $\hat{\Gamma}$ spanned by the top-$k$ eigenvectors of $A$ defined in (1). For $\ell = 1, \ldots, d$, we define $\sigma_\ell \geq 0$ such that $\sigma_\ell^2$ is the $\ell$-th largest

eigenvalue of $\sum_{i=1}^{n} w_i \mu_i \mu_i^\mathsf{T}$. Since $\mu_1, \ldots, \mu_n$ share a $k$-dimensional subspace, $\sigma_\ell = 0$ for $\ell > k$. We prove the following general theorem on the error guarantee of our estimator:

**Theorem 3.1.** *Define $\xi^2 = \|\sum_{i=1}^{n} w_i^2 \mu_i \mu_i^\mathsf{T} \eta_i^2 / m_i\|$ and let $\theta$ denote the maximum principal angle between our estimator $\hat{\Gamma}$ and the true subspace $\Gamma$ spanned by $\mu_1, \ldots, \mu_n$. For any $\delta \in (0, 1/2)$, with probability at least $1 - \delta$,*

$$\sin \theta = O\left(\sigma_k^{-2} \sqrt{(d + \log(1/\delta))\left(\xi^2 + \sum_{i=1}^{n} \frac{w_i^2 \eta_i^4}{m_i^2}\right)} + \sigma_k^{-2}(d + \log(1/\delta)) \max_i \frac{w_i \eta_i^2}{m_i}\right). \quad (6)$$

We can simplify the bound in Theorem 3.1 by considering special cases:

**Corollary 3.2.** *Assume $\max\{\eta_1/\sqrt{m_1}, \ldots, \eta_n/\sqrt{m_n}\} = t$ and we choose $w_1 = \cdots = w_n = 1/n$. For any $\delta \in (0, 1/2)$, with probability at least $1 - \delta$,*

$$\sin \theta = O\left(\frac{t\sigma_1 + t^2}{\sigma_k^2} \sqrt{\frac{d + \log(1/\delta)}{n}}\right). \quad (7)$$

*In particular, when $\eta_1 = \cdots = \eta_n = \eta$, and $m_1 = \cdots = m_n = m$, error bound (7) becomes*

$$\sin \theta = O\left(\left(\frac{\eta \sigma_1}{\sigma_k^2 \sqrt{m}} + \frac{\eta^2}{\sigma_k^2 m}\right) \sqrt{\frac{d + \log(1/\delta)}{n}}\right).$$

We defer the complete proof of Theorem 3.1 and Corollary 3.2 to Appendices F and G. Our proof is based on the Davis-Kahan $\sin \theta$ theorem (Theorem 2.1). Since $\sigma_{k+1}^2 = 0$, Theorem 2.1 implies

$$\sin \theta \leq \frac{2\|A - \sum_{i=1}^{n} w_i \mu_i \mu_i^\mathsf{T}\|}{\sigma_k^2}. \quad (8)$$

This reduces our goal to proving an upper bound on the spectral norm of $A - \sum_{i=1}^{n} w_i \mu_i \mu_i^\mathsf{T}$. Since for distinct $j_1$ and $j_2$ in $\{1, \ldots, m_i\}$ we have $\mathbb{E}[x_{ij_1} x_{ij_2}^\mathsf{T}] = \mu_i \mu_i^\mathsf{T}$, our construction of $A$ in (1) guarantees $\mathbb{E}[A] = \sum_{i=1}^{n} w_i \mu_i \mu_i^\mathsf{T}$. Therefore, our goal becomes controlling the deviation of $A$ from its expectation, and we achieve this goal using techniques for matrix concentration inequalities.

## 3.2 Sub-Gaussian User-Specific Means

We apply our error upper bound in Theorem 3.1 to the case where $\mu_1, \ldots, \mu_n \in \mathbb{R}^d$ are drawn iid from $N(0, \sigma^2 UU^\mathsf{T})$ for an unknown $U \in \mathbb{O}_{d,k}$. We still assume that each data point $x_{ij} \in \mathbb{R}^d$ is generated by adding a noise vector $z_{ij} \in \mathbb{R}^d$ to the user-specific mean $\mu_i$ as in (5). We do *not* assume that the noise vectors $(z_{ij})_{1 \leq i \leq n, 1 \leq j \leq m_i}$ are independent of the user-specific means $(\mu_i)_{1 \leq i \leq n}$, but we assume that when conditioned on $(\mu_i)_{1 \leq i \leq n}$, every noise vector $z_{ij}$ independently follows a distribution with mean zero and sub-Gaussian constant $\eta_i$. We use the same estimator $\hat{\Gamma}$ as before: $\hat{\Gamma}$ is the subspace spanned by the top-$k$ eigenvectors of $A$ defined in (1). We determine the optimal weights $w_1, \ldots, w_n$ in (1) as long as $m_1, \ldots, m_n$ and $\eta_1, \ldots, \eta_n$ satisfy a mild assumption (Assumption 3.2), achieving an error upper bound in Theorem 3.4. In the next subsection, we prove an error lower bound (Theorem 3.7) that matches our upper bound (Theorem 3.4) up to a constant factor, assuming $d \geq (1 + \Omega(1))k$ and $\delta = \Theta(1)$.

We prove our error upper bound in a slightly more general setting than $\mu_1, \ldots, \mu_n$ drawn iid from $N(0, \sigma^2 UU^\mathsf{T})$. Specifically, we make the following assumption on the distribution of $\mu_1, \ldots, \mu_n$:

**Assumption 3.1.** *The user-specific means $\mu_1, \ldots, \mu_n \in \mathbb{R}^d$ are mean-zero independent random vectors supported on an unknown $k$-dimensional subspace $\Gamma$. Moreover, for a parameter $\sigma > 0$, for every $i = 1, \ldots, n$, $\mu_i$ has sub-Gaussian constant $O(\sigma)$, and the $k$-th largest eigenvalue of $\mathbb{E}[\mu_i \mu_i^\mathsf{T}]$ is at least $\sigma^2$.*

Under this assumption, we have the following lower bound on the $\sigma_k^2$ in Theorem 3.1 (see Appendix H for proof):

**Claim 3.3.** *Under Assumption 3.1, let $w_1, \dots, w_n \in \mathbb{R}_{\geq 0}$ be user weights satisfying $w_1 + \cdots + w_n = 1$ and $\sigma_k^2$ be the $k$-th largest eigenvalue of $\sum_{i=1}^n w_i \mu_i \mu_i^\top$. There exists an absolute constant $C_* > 1$ such that for any $\delta \in (0, 1/2)$, as long as $\max_{1 \leq i \leq n} w_i \leq 1/C_*(k + \log(1/\delta))$, then $\sigma_k^2 \geq \sigma^2/2$ with probability at least $1 - \delta/2$.*

The following definition is important for us to choose the weights $w_1, \dots, w_n$ in (1) optimally:

**Definition 3.1.** *Define $\gamma_i = \left( \frac{\eta_i^2}{\sigma^2 m_i} + \frac{\eta_i^4}{\sigma^4 m_i^2} \right)^{-1}$ and assume w.l.o.g. that $\gamma_1 \geq \cdots \geq \gamma_n$. Define $\gamma_i' = \gamma_i$ if $i \geq k$, and $\gamma_i' = \gamma_k$ if $i < k$.*

Intuitively, we can view $\gamma_i$ as measuring the "amount of information" provided by the data points from user $i$. This is consistent with the fact that $\gamma_i$ increases as the number $m_i$ of data points from user $i$ increases, and $\gamma_i$ decreases as the noise magnitude $\eta_i$ from user $i$ increases. With the users sorted so that $\gamma_1 \geq \cdots \geq \gamma_n$, the quantity $\gamma_i'$ is then defined to be $\gamma_k$ for the $k$ most "informative" users $i = 1, \dots, k$, and $\gamma_i' = \gamma_i$ for other users. We make the following mild assumption on $\gamma_i'$ under which we achieve optimal estimation error:

**Assumption 3.2.** $\sum_{i=1}^n \gamma_i' \geq C_*(k + \log(1/\delta))\gamma_1'$ *for $C_*$ defined in Claim 3.3.*

By the definition of $\gamma_i'$, it is easy to show that Assumption 3.2 is equivalent to $\sum_{i=k+1}^n \gamma_i \geq ((C_* - 1)k + C_* \log(1/\delta))\gamma_k$. Therefore, if we view $\gamma_i$ as the "amount of information" from user $i$, Assumption 3.2 intuitively requires that a significant contribution to the total "information" comes from outside the $k$ most "informative" users. This assumption allows us to avoid the case where we only have exactly $n = k$ users: in that case, we would have $\sigma_k^2 \approx \sigma^2/k^2$ for uniform weights $w_1 = \cdots = w_n$ (see [Rudelson and Vershynin, 2008] and references therein), as opposed to the desired $\sigma_k^2 \geq \sigma^2/2$ in Claim 3.3.

Assumption 3.2 is a mild assumption. For example, when $\gamma_k = \cdots = \gamma_n$, Assumption 3.2 holds as long as $n \geq C_*(k + \log(1/\delta))$. Also, since $\gamma_1' = \cdots = \gamma_k' \geq \gamma_{k+1}' \geq \cdots \geq \gamma_n' \geq 0$, it trivially holds that $\sum_{i=1}^n \gamma_i' \geq k\gamma_1'$. Assumption 3.2 is relatively mild when compared to this trivial inequality.

Under Assumption 3.2, we show that it is optimal to choose the weights $w_1, \dots, w_n$ as

$$w_i = \frac{\gamma_i'}{\sum_{\ell=1}^n \gamma_\ell'}. \tag{9}$$

Specifically, if we plug (9) into Theorem 3.1 and bound $\xi$ and $\sigma_k$ based on the distribution of $\mu_1, \dots, \mu_n$, we get the following error upper bound which matches our lower bound (Theorem 3.7) in Section 3.3. We defer its proof to Appendix I.

**Theorem 3.4.** *Under Assumptions 3.1 and 3.2, if we choose $w_1, \dots, w_n$ as in (9) and define $\theta = \angle(\Gamma, \hat{\Gamma})$, for $\delta \in (0, 1/2)$, with probability at least $1 - \delta$,*

$$\sin \theta \leq O\left( \sqrt{\frac{d + \log(1/\delta)}{\sum_{i=1}^n \gamma_i'}} \right). \tag{10}$$

For comparison, consider the setting when $\sigma = \eta_i = 1$ for every $i = 1, \dots, n$. The result then says that $\sin \theta$ is bounded by approximately $\sqrt{\frac{d}{\sum_{i=1}^n m_i}}$. This is the same rate as we would get if we have $\sum_{i=1}^n m_i$ users each contributing a single independent data point with homogeneous spherical noise. Thus as long as the data points are not too concentrated on fewer than $k$ users, the heterogeneity comes at no additional cost.

### 3.3 Lower Bound

We prove a lower bound matching the upper bound in Theorem 3.4 up to constant in the setting where $\delta = \Theta(1)$, $d \geq (1 + \Omega(1))k$.

For every positive integer $d$, there is a natural "uniform" distribution over $\mathbb{O}_d$ given by Haar's theorem [Haar, 1933] (see e.g. [Diestel and Spalsbury, 2014] for a textbook). We denote this distribution by $\mathsf{Haar}(\mathbb{O}_d)$. A random matrix $A$ drawn from $\mathsf{Haar}(\mathbb{O}_d)$ has the following invariance property: for any deterministic matrix $B \in \mathbb{O}_d$, the random matrices $A$, $AB$ and $BA$ all have the same distribution.

For an integer $k \leq d$, we can construct a random matrix $A_1 \in \mathbb{O}_{d,k}$ by first drawing $A \in \mathbb{R}^{d \times d}$ from $\mathsf{Haar}(\mathbb{O}_d)$ and then take the first $k$ columns of $A$. We denote the distribution of $A_1$ by $\mathsf{Haar}(\mathbb{O}_{d,k})$. The invariance property of $\mathsf{Haar}(\mathbb{O}_d)$ immediately implies the following claims:

**Claim 3.5.** *Let $A \in \mathbb{O}_d$ be a random matrix drawn from $\mathsf{Haar}(\mathbb{O}_d)$ and let $B \in \mathbb{O}_{d,k}$ be a fixed matrix. Then $AB$ distributes as $\mathsf{Haar}(\mathbb{O}_{d,k})$.*

*Proof.* The matrix $B$ can be written as the first $k$ columns of a matrix $C \in \mathbb{O}_d$. Now $AB$ is the first $k$ columns of $AC$, where $AC$ distributes as $\mathsf{Haar}(\mathbb{O}_d)$ by the invariance property. This implies that $AB$ distributes as $\mathsf{Haar}(\mathbb{O}_{d,k})$. □

**Claim 3.6.** *Let $B \in \mathbb{O}_{d,k}$ be a random matrix. Assume for every fixed matrix $A \in \mathbb{O}_d$, the random matrices $B$ and $AB$ have the same distribution. Then $B \sim \mathsf{Haar}(\mathbb{O}_{d,k})$.*

*Proof.* If we draw $A$ independently from $\mathsf{Haar}(\mathbb{O}_d)$, the random matrices $B$ and $AB$ still have the same distribution. By Claim 3.5, $AB$ distributes as $\mathsf{Haar}(\mathbb{O}_{d,k})$, so $B$ must also distribute as $\mathsf{Haar}(\mathbb{O}_{d,k})$. □

With the definition of $\mathsf{Haar}(\mathbb{O}_{d,k})$, we state our lower bound in the following theorem:

**Theorem 3.7.** *Let $k, d, n$ be positive integers satisfying $k < d$ and $k \leq n$. Let $m_1, \ldots, m_n$ be positive integers and $\sigma, \eta_1, \ldots, \eta_n$ be positive real numbers. Suppose we draw $U \in \mathbb{O}_{d,k}$ from $\mathsf{Haar}(\mathbb{O}_{d,k})$ and then draw $\mu_1, \ldots, \mu_n$ independently from $N(0, \sigma^2 UU^\mathsf{T})$. For every $i = 1, \ldots, n$, we draw $m_i$ data points $x_{ij}$ for $j = 1, \ldots, m_i$ as $x_{ij} = \mu_i + z_{ij}$, where each $z_{ij}$ is drawn independently from the spherical Gaussian $N(0, \eta_i^2 I)$. Let $\hat{\Gamma}$ be any estimator mapping $(x_{ij})_{1 \leq i \leq n, 1 \leq j \leq m_i}$ to a (possibly randomized) $k$-dimensional subspace of $\mathbb{R}^d$. Let $\theta$ denote the maximum principal angle between $\hat{\Gamma}((x_{ij})_{1 \leq i \leq n, 1 \leq j \leq m_i})$ and the true subspace $\Gamma = \mathsf{col}(U)$. If real numbers $t \geq 0$ and $\delta \in [0, 1/2)$ satisfy $\Pr[\sin \theta \leq t] \geq 1 - \delta$, then*

$$t \geq \Omega \left( \min \left\{ 1, \sqrt{\frac{(d-k)(1-\delta)}{\sum_{i=k}^n \gamma_i}} \right\} \right), \tag{11}$$

*where $\gamma_1, \ldots, \gamma_n$ are defined in Definition 3.1.*

Note that $\gamma_i' = \gamma_i$ for $i \geq k$, so our upper bound in (10) matches the lower bound (11) up to a constant factor assuming $\delta = \Theta(1)$ and $d \geq (1 + \Omega(1))k$.

We use the local Fano method to prove the lower bound using the technical lemmas in Appendix D. In particular, we reduce our goal to proving an upper bound on the KL divergence between Gaussian distributions whose covariance matrices are defined based on matrices $U, \hat{U} \in \mathbb{O}_{d,k}$ with $\|UU^\mathsf{T} - \hat{U}\hat{U}^\mathsf{T}\|_F$ bounded. We prove the following lemma in Appendix J that upper bounds the KL divergence using $\|UU^\mathsf{T} - \hat{U}\hat{U}^\mathsf{T}\|_F$:

**Lemma 3.8.** *For $\sigma \in \mathbb{R}_{\geq 0}, \eta \in \mathbb{R}_{>0}, U, \hat{U} \in \mathbb{O}_{d,k}$, define $\Sigma = \sigma^2 UU^\mathsf{T} + \eta^2 I$ and $\hat{\Sigma} = \sigma^2 \hat{U}\hat{U}^\mathsf{T} + \eta^2 I$. Then,*

$$D_{\mathrm{kl}}(N(0, \hat{\Sigma}) \| N(0, \Sigma)) = \frac{\sigma^4 \|UU^\mathsf{T} - \hat{U}\hat{U}^\mathsf{T}\|_F^2}{4(\sigma^2\eta^2 + \eta^4)}.$$

Lemma 3.8 and the results in Appendix D allow us to prove a version of (11) in which the sum in the demoninator is over $i = 1, \ldots, n$. This, however, is weaker and less useful than (11) in which the sum in the denominator is over $i = k, k + 1, \ldots, n$. To prove Theorem 3.7, we extract a hard distribution in which the data points from users $1, \ldots, k - 1$ are "useless" in terms of subspace recovery.

Let $\Gamma_1$ be the $(k-1)$-dimensional subspace spanned by $\mu_1, \ldots, \mu_{k-1}$. We let $v_1, \ldots, v_{k-1}$ be a random orthonormal basis of $\Gamma_1$, and we append another vector $v_k \in \Gamma$ to form an orthonormal basis $v_1, \ldots, v_k$ of $\Gamma$. We define $V_1 = [v_1 \ \cdots \ v_{k-1}] \in \mathbb{O}_{d,k-1}$ and $V = [v_1 \ \cdots \ v_k] \in \mathbb{O}_{d,k}$. In Figure 1 we show a graphical model demonstrating the dependency among the random objects we defined.

Let us focus on the joint distribution of $(V_1, V, (\mu_1, \ldots, \mu_{k-1}))$. By the invariance property, for any matrices $\tilde{V}_1 \in \mathbb{O}_{d,k-1}, \tilde{V} \in \mathbb{O}_{d,k}$, measurable set $S \subseteq (\mathbb{R}^d)^{k-1}$, and orthogonal matrix $G \in \mathbb{O}_d$,

$$\Pr[(\mu_1, \ldots, \mu_{k-1}) \in S | V = \tilde{V}, V_1 = \tilde{V}_1] = \Pr[(\mu_1, \ldots, \mu_{k-1}) \in S_G | V = G\tilde{V}, V_1 = G\tilde{V}_1],$$

Figure 1: Graphical model A.

where $S_G = \{(G\tilde{\mu}_1, \ldots, G\tilde{\mu}_{k-1}) : (\tilde{\mu}_1, \ldots, \tilde{\mu}_{k-1}) \in S\}$. For any $\tilde{V}, \tilde{V}' \in \mathbb{O}_{d,k}$ whose first $k-1$ columns are both $\tilde{V}_1$, there exists $G \in \mathbb{O}_d$ such that $\tilde{V}' = G\tilde{V}$ and thus $\tilde{V}_1 = G\tilde{V}_1$. This implies that for any $\tilde{\mu} \in \mathsf{col}(\tilde{V}_1)$, we have $G\tilde{\mu} = \tilde{\mu}$, and thus $(S \cap \mathsf{col}(\tilde{V}_1)^{k-1})_G = S \cap \mathsf{col}(\tilde{V}_1)^{k-1}$ for any measurable $S \subseteq (\mathbb{R}^d)^{k-1}$. Here, $\mathsf{col}(\tilde{V}_1)^{k-1} = \{(\tilde{\mu}_1, \ldots, \tilde{\mu}_{k-1}) : \tilde{\mu}_i \in \mathsf{col}(\tilde{V}_1) \text{ for } i = 1, \ldots, k-1\} \subseteq (\mathbb{R}^d)^{k-1}$. When conditioned on $V_1 = \tilde{V}_1$, for every $i = 1, \ldots, k-1$ we have $\mu_i \in \Gamma_1 = \mathsf{col}(V_1) = \mathsf{col}(\tilde{V}_1)$, which implies that $(\mu_1, \ldots, \mu_{k-1}) \in \mathsf{col}(\tilde{V}_1)^{k-1}$. Therefore,

$$\Pr[(\mu_1, \ldots, \mu_{k-1}) \in S | V = \tilde{V}, V_1 = \tilde{V}_1]$$
$$= \Pr[(\mu_1, \ldots, \mu_{k-1}) \in S \cap \mathsf{col}(\tilde{V}_1)^{k-1} | V = \tilde{V}, V_1 = \tilde{V}_1]$$
$$= \Pr[(\mu_1, \ldots, \mu_{k-1}) \in (S \cap \mathsf{col}(\tilde{V}_1)^{k-1})_G | V = G\tilde{V}, V_1 = G\tilde{V}_1]$$
$$= \Pr[(\mu_1, \ldots, \mu_{k-1}) \in S \cap \mathsf{col}(\tilde{V}_1)^{k-1} | V = \tilde{V}', V_1 = \tilde{V}_1]$$
$$= \Pr[(\mu_1, \ldots, \mu_{k-1}) \in S | V = \tilde{V}', V_1 = \tilde{V}_1].$$

This implies that $(\mu_1, \ldots, \mu_{k-1})$ and $V$ are conditionally independent given $V_1$. Therefore, the joint distribution of $(V_1, V, (\mu_1, \ldots, \mu_{k-1}))$ can be formed by first drawing $V$ and $V_1$, and then drawing $\mu_1, \ldots, \mu_{k-1}$ based only on $V_1$ and not on $V$. Since $\mu_k, \ldots, \mu_n$ are drawn iid from $N(0, \sigma^2 UU^{\mathsf{T}}) = N(0, \sigma^2 VV^{\mathsf{T}})$, we have the graphical model shown in Figure 2.

Figure 2: Graphical model B.

By Claim 3.6, the marginal distribution of $V$ is $\mathsf{Haar}(\mathbb{O}_{d,k})$. By Claim 3.5, we can implement this distribution by first drawing $W \sim \mathsf{Haar}(\mathbb{O}_d)$ and then drawing $E$ independently from any distribution over $\mathbb{O}_{d,k}$ and let $V = WE$. We choose the distribution of $E$ later, where we ensure that the first $k-1$ columns of $E$ is always $\begin{bmatrix} I_{k-1} \\ 0 \end{bmatrix}$. This guarantees that the first $k-1$ columns of $W$ and $V$ are the same, and thus $V_1$ is exactly the first $k-1$ columns of $W$, resulting in the graphical model shown in Figure 3.

Figure 3: Graphical model C.

Note that in Figure 3 there is no directed path from $E$ to $(\mu_1, \ldots, \mu_{k-1})$. Intuitively, this means that knowing $(\mu_1, \ldots, \mu_{k-1})$ gives us no information about $E$. Now by choosing the distribution of $E$ appropriately, we can prove (11) in which the denominator does not contain $\gamma_1, \ldots, \gamma_{k-1}$. We defer the complete proof of Theorem 3.7 to Appendix K.

## 4 Linear Models

In the linear models setting, the data distribution of user $i$ is parameterized by an unknown vector $\beta_i \in \mathbb{R}^d$. As before, we assume that the vectors $\beta_1, \ldots, \beta_n$ from the $n$ users lie in an unknown $k$-dimensional subspace $\Gamma$. Our goal is to recover the subspace using the following data. For every

$i = 1, \ldots, n$, we have $m_i$ data points from user $i$: $(x_{i1}, y_{i1}), \ldots, (x_{im_i}, y_{im_i}) \in \mathbb{R}^d \times \mathbb{R}$. For every $j = 1, \ldots, m_i$, we assume the *measurement* $x_{ij} \in \mathbb{R}^d$ is a random vector drawn independently from an $O(1)$-sub-Gaussian distribution with zero mean and identity covariance matrix. The measurement *outcome* $y_{ij}$ is determined by $y_{ij} = x_{ij}^\mathsf{T} \beta_i + z_{ij}$, where the random *noise* $z_{ij} \in \mathbb{R}$ can depend on the measurements $x_{i1}, \ldots, x_{im_i}$. When conditioned on $x_{i1}, \ldots, x_{im_i}$, we assume every $z_{ij}$ for $j = 1, \ldots, m_i$ is independently drawn from an $\eta_i$-sub-Gaussian distribution with zero mean, but we do *not* assume that the conditional distribution of $z_{ij}$ is the same for every $j = 1, \ldots, m_i$. The (in)dependence among $x_{ij}$ and $z_{ij}$ for $i = 1, \ldots, n$ and $j = 1, \ldots, m_i$ can be summarized by the example graphical model in Figure 4.

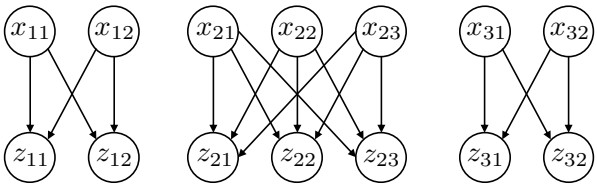

Figure 4: An example for $n = 3, m_1 = 2, m_2 = 3, m_3 = 2$.

Since we allow the noise $z_{ij}$ to depend on the measurements $x_{ij}$, it is information-theoretically impossible to recover the subspace if we only have one data point from every user. Consider the scenario where every $\beta_i$ is drawn independently from $N(0, \sigma^2 uu^\mathsf{T})$ for an unknown unit vector $u \in \mathbb{R}^d$ and every $x_{ij}$ is drawn independently and uniformly from $\{-1, 1\}^d$. If we set $z_{ij}$ to be $z_{ij} = x_{ij}^\mathsf{T} \nu_{ij}$ where $\nu_{ij}$ is independently drawn from $N(0, \sigma^2(I - uu^\mathsf{T}))$, then every $y_{ij}$ satisfies $y_{ij} = x_{ij}^\mathsf{T}(\beta_i + \nu_{ij})$ where $\beta_i + \nu_{ij}$ distributes as $N(0, \sigma^2 I)$ independently from $x_{ij}$. This implies that the joint distribution of $((x_{i1}, y_{i1}))_{i=1,\ldots,n}$ does not change with $u$, i.e., we get no information about $u$ from one data point per user.

Thus, we assume $m_i \geq 2$ for every user $i$. In this case, we achieve error upper bounds that match the ones in [Tripuraneni et al., 2021] despite our relaxed assumptions on the noise. Our estimator is the subspace $\hat{\Gamma}$ spanned by the top-$k$ eigenvectors of $A$ defined in (3). We defer the analysis of our estimator to Appendix L.

## Acknowledgments and Disclosure of Funding

Part of this work was performed while LH was interning at Apple. LH is also supported by Omer Reingold's NSF Award IIS-1908774, Omer Reingold's Simons Foundation Investigators Award 689988, and Moses Charikar's Simons Foundation Investigators Award.

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
