## A  Basics about Principal Angles

We provide a formal definition of the principal angles introduced by Jordan [1875] and briefly discuss some basic properties of this notion (see e.g. [Stewart and Sun, 1990] for a textbook).

Let $U, \hat{U} \in \mathbb{O}_d$ be two orthogonal matrices. Suppose the columns of $U$ and $\hat{U}$ are partitioned as $U = [U_1 \ U_2], \hat{U} = [\hat{U}_1 \ \hat{U}_2]$ where $U_1 \in \mathbb{O}_{d,k}$ and $\hat{U}_1 \in \mathbb{O}_{d,\hat{k}}$ for integers $k, \hat{k}$ satisfying $0 < k \le \hat{k} < d$. Starting from a singular value decomposition of $U_1^\mathsf{T} \hat{U}_1$ as $R_1^\mathsf{T} \Sigma \hat{R}_1$, one can find an integer $\ell \in \mathbb{Z}_{\ge 0}$ and angles $0 < \theta_1 \le \cdots \le \theta_\ell \le \pi/2$ such that there exist orthogonal matrices $P, \hat{P} \in \mathbb{O}_d, R_1 \in \mathbb{O}_k, R_2 \in \mathbb{O}_{d-k}, \hat{R}_1 \in \mathbb{O}_{\hat{k}}, \hat{R}_2 \in \mathbb{O}_{d-\hat{k}}$ satisfying

$$U = P \begin{bmatrix} R_1 & 0 \\ 0 & R_2 \end{bmatrix}, \quad \hat{U} = \hat{P} \begin{bmatrix} \hat{R}_1 & 0 \\ 0 & \hat{R}_2 \end{bmatrix}, \quad \text{and} \tag{12}$$

$$P^\mathsf{T} \hat{P} = (\hat{P}^\mathsf{T} P)^\mathsf{T} = \begin{bmatrix} I_{k-\ell} & 0 & 0 & 0 & 0 \\ 0 & \cos\Theta & 0 & \sin\Theta & 0 \\ 0 & 0 & I_{\hat{k}-k} & 0 & 0 \\ 0 & -\sin\Theta & 0 & \cos\Theta & 0 \\ 0 & 0 & 0 & 0 & I_{d-\hat{k}-\ell} \end{bmatrix}, \tag{13}$$

where $\cos\Theta = \operatorname{diag}(\cos\theta_1, \ldots, \cos\theta_\ell)$ and $\sin\Theta = \operatorname{diag}(\sin\theta_1, \ldots, \sin\theta_\ell)$. It is easy to see that $\cos\theta_1, \ldots, \cos\theta_\ell$ are exactly the singular values of $U_1^\mathsf{T} \hat{U}_1$ that are not equal to 1, so $\ell, \theta_1, \ldots, \theta_\ell$ are unique. Let $\Gamma$ (resp. $\hat{\Gamma}$) be the $k$-dimensional (resp. $\hat{k}$-dimensional) linear subspace spanned by the columns of $U_1$ (resp. $\hat{U}_1$). The (non-zero) *principal angles* between $\Gamma$ and $\hat{\Gamma}$ are defined to be $\theta_1, \ldots, \theta_\ell$. It is not hard to see that the principal angles depend only on the subspaces $\Gamma, \hat{\Gamma}$ and not on the choices of $U$ and $\hat{U}$. The *maximum principal angle* between $\Gamma$ and $\hat{\Gamma}$, denoted by $\angle(\Gamma, \hat{\Gamma})$ or $\angle(U_1, \hat{U}_1)$, is defined to be $\theta_\ell$ (or 0 when $\ell = 0$). When $k = \hat{k}$, using (12) and (13), it is not hard to show that the non-zero eigenvalues of $U_1 U_1^\mathsf{T} - \hat{U}_1 \hat{U}_1^\mathsf{T}$ are $\pm \sin\theta_1, \ldots, \pm \sin\theta_\ell$. This implies $\|U_1 U_1^\mathsf{T} - \hat{U}_1 \hat{U}_1^\mathsf{T}\| = \sin\angle(\Gamma, \hat{\Gamma})$ and $\|U_1 U_1^\mathsf{T} - \hat{U}_1 \hat{U}_1^\mathsf{T}\|_F = \sqrt{2 \sum_{i=1}^\ell \sin^2 \theta_i}$. In particular, $\sin\angle(\Gamma, \hat{\Gamma})$ is a natural metric between $k$-dimensional subspaces.

## B  Basic Facts and Concentration Inequalities

**Lemma B.1** ([See Vershynin, 2018, Proposition 2.6.1]). *If $X_1, \ldots, X_n \in \mathbb{R}$ are independent random variables with sub-Gaussian constants $b_1, \ldots, b_n \in \mathbb{R}_{\ge 0}$ respectively, then $\sum_{i=1}^n X_i$ has sub-Gaussian constant $O\left(\sqrt{\sum_{i=1}^n b_i^2}\right)$.*

**Lemma B.2** ([See Vershynin, 2018, Lemma 2.7.6]). *If $X \in \mathbb{R}$ is a mean-zero random variable with sub-Gaussian constant $b \in \mathbb{R}_{\ge 0}$, then $X^2$ has sub-exponential constant $O(b^2)$.*

**Lemma B.3** ([See Vershynin, 2018, Theorem 2.6.2]). *Let $x_1, \ldots, x_n \in \mathbb{R}$ be independent random variables. For every $i = 1, \ldots, n$, assume $x_i$ has mean zero and sub-Gaussian constant $b_i \in \mathbb{R}_{\ge 0}$. Then for every $\delta \in (0, 1/2)$, with probability at least $1 - \delta$,*

$$\left| \sum_{i=1}^n x_i \right| \le O\left( \sqrt{\log(1/\delta) \sum_{i=1}^n b_i^2} \right).$$

**Lemma B.4** ([See Vershynin, 2018, Theorem 2.8.1]). *Let $x_1, \ldots, x_n \in \mathbb{R}$ be independent random variables. For every $i = 1, \ldots, n$, assume $x_i$ has mean zero and sub-exponential constant $b_i \in \mathbb{R}_{\ge 0}$. Then for every $\delta \in (0, 1/2)$, with probability at least $1 - \delta$,*

$$\left| \sum_{i=1}^n x_i \right| \le O\left( \sqrt{\log(1/\delta) \sum_{i=1}^n b_i^2} + \log(1/\delta) \max_{i=1,\ldots,n} b_i \right).$$

## C  Vector and Matrix Concentration Inequalities

**Lemma C.1** ([See Vershynin, 2018, Corollary 4.2.13 and Exercise 4.4.3]). *There exists an absolute constant $C > 1$ and a set $\mathbb{O}'_d \subseteq \mathbb{O}_{d,1}$ for every positive integer $d$ such that*

1. $|O'_d| \le 2^{Cd}$ for every $d \in \mathbb{Z}_{>0}$.

2. for every $d \in \mathbb{Z}_{>0}$ and every symmetric matrix $A \in \mathbb{R}^{d \times d}$,
$$\|A\| \le C \sup_{u \in O'} |u^\mathsf{T} A u|.$$

3. for every $d_1, d_2 \in \mathbb{Z}_{>0}$ and every matrix $A \in \mathbb{R}^{d_1 \times d_2}$,
$$\|A\| \le C \sup_{u_1 \in O'_{d_1}} \sup_{u_2 \in O'_{d_2}} |u_1^\mathsf{T} A u_2|.$$

**Lemma C.2.** *Suppose $x_1, \ldots, x_n \in \mathbb{R}^d$ are independent random vectors, and each $x_i$ is mean-zero and 1-sub-Gaussian. Suppose $w_1, \ldots, w_n \in \mathbb{R}$ are fixed real numbers. Define $E = \sum_{i=1}^n w_i x_i x_i^\mathsf{T}$ and $\bar{E} = \mathbb{E}[E]$. Then for any $\delta \in (0, 1/2)$, with probability at least $1 - \delta$,*
$$\|E - \bar{E}\| \le O\left( \sqrt{(d + \log(1/\delta)) \sum_{i=1}^n w_i^2} + (d + \log(1/\delta)) \max_i |w_i| \right).$$

*Proof.* Let $O'$ denote the set $O'_d \subseteq \mathbb{O}_{d,1}$ guaranteed by Lemma C.1. For any fixed $u \in O'$, by Lemma B.2, $(x_i^\mathsf{T} u)^2$ has sub-exponential constant $O(1)$, and thus $w_i(x_i^\mathsf{T} u)^2$ has sub-exponential constant $|w_i|$. Define $\delta' := \delta/|O'|$. By Lemma B.4, with probability at least $1 - \delta'$,
$$\left| \sum_{i=1}^n w_i(x_i^\mathsf{T} u)^2 - \mathbb{E}\left[ \sum_{i=1}^n w_i(x_i^\mathsf{T} u)^2 \right] \right| \le O\left( \sqrt{\log(1/\delta') \sum_{i=1}^n w_i^2} + \log(1/\delta') \max_i |w_i| \right). \quad (14)$$

By a union bound, with probability at least $1 - \delta$, the above inequality holds for every $u \in O'$. By the definition of $O'$,
$$\|E - \bar{E}\| \le O\left( \sup_{u \in O'} |u(E - \bar{E})u| \right) = O\left( \sup_{u \in O'} \left| \sum_{i=1}^n w_i(x_i^\mathsf{T} u)^2 - \mathbb{E}\left[ \sum_{i=1}^n w_i(x_i^\mathsf{T} u)^2 \right] \right| \right). \quad (15)$$

Combining (14) and (15) and noting that $\log(1/\delta') = O(d + \log(1/\delta))$ completes the proof. $\qquad \square$

**Lemma C.3.** *Suppose $x_1, \ldots, x_n \in \mathbb{R}^d$ are independent random vectors, and each $x_i$ is mean-zero and 1-sub-Gaussian. Suppose $b_1, \ldots, b_n \in \mathbb{R}$ are fixed real numbers. Then for any $\delta \in (0, 1/2)$, with probability at least $1 - \delta$,*
$$\| [x_1 \ \cdots \ x_n] \operatorname{diag}(b_1, \ldots, b_n) \| \le O\left( \sqrt{\sum_{i=1}^n b_i^2 + (d + \log(1/\delta)) \max_i b_i^2} \right).$$

*Proof.* Define $F = [x_1 \ \cdots \ x_n] \operatorname{diag}(b_1, \ldots, b_n)$ and $E = FF^\mathsf{T} = \sum_{i=1}^n b_i^2 x_i x_i^\mathsf{T}$. By Lemma C.2, with probability at least $1 - \delta$,
$$\|E - \mathbb{E}[E]\| \le O\left( \sqrt{(d + \log(1/\delta)) \sum_{i=1}^n b_i^4} + (d + \log(1/\delta)) \max_i b_i^2 \right). \quad (16)$$

For every unit vector $u \in \mathbb{R}^d$, $x_i^\mathsf{T} u$ is 1-sub-Gaussian, and thus $\mathbb{E}[(x_i^\mathsf{T} u)^2] \le O(1)$. This implies that $u \mathbb{E}[x_i x_i^\mathsf{T}] u \le O(1)$ for every unit vector $u \in \mathbb{R}^d$ and thus $\|\mathbb{E}[x_i x_i^\mathsf{T}]\| \le O(1)$. Now we have
$$\|\mathbb{E}[E]\| \le \sum_{i=1}^n b_i^2 \|x_i x_i^\mathsf{T}\| \le O\left( \sum_{i=1}^n b_i^2 \right). \quad (17)$$

Combining (16) and (17) and using the fact that $\sum_{i=1}^n b_i^4 \le \left( \sum_{i=1}^n b_i^2 \right) \max_i b_i^2$, with probability at least $1 - \delta$,
$$\|E\| \le \|E - \mathbb{E}[E]\| + \|\mathbb{E}[E]\| \le O\left( \sum_{i=1}^n b_i^2 + (d + \log(1/\delta)) \max_i b_i^2 \right).$$

The lemma is proved by $\|F\| = \sqrt{\|E\|}$. $\qquad \square$

**Lemma C.4.** *Let $X \in \mathbb{R}^{d \times m}$ be a fixed matrix and let $v \in \mathbb{R}^m$ be a mean zero random vector with sub-Gaussian constant 1. For $\delta \in (0, 1/2)$, with probability at least $1 - \delta$,*

$$\|Xv\|_2 \leq O(\|X\|\sqrt{r + \log(1/\delta)}),$$

*where $r$ is the rank of $X$.*

*Proof.* By the singular value decomposition, there exists $P \in \mathbb{O}_d$ such that all but the first $r$ rows of $PX$ are zeros. Let $Y \in \mathbb{R}^{r \times m}$ denote the first $r$ rows of $PX$. We have

$$\|Xv\|_2 = \|PXv\|_2 = \|Yv\|_2. \tag{18}$$

Let $O'$ be the $O'_r$ in Lemma C.1. For every $u \in O'$, $u^\mathsf{T} Y v$ has sub-Gaussian constant $O(\|u^\mathsf{T} Y\|_2)$. By Lemma B.3, for $\delta' = \delta/|O'|$, with probability at least $1 - \delta'$,

$$|u^\mathsf{T} Y v| \leq O\left(\|u^\mathsf{T} Y\|_2 \sqrt{\log(1/\delta')}\right) \leq O\left(\|Y\| \sqrt{\log(1/\delta')}\right).$$

By a union bound, with probability at least $1 - \delta$,

$$\sup_{u \in O'} |u^\mathsf{T} Y v| \leq O\left(\|u^\mathsf{T} Y\|_2 \sqrt{\log(1/\delta')}\right) \leq O\left(\|Y\| \sqrt{\log(1/\delta')}\right). \tag{19}$$

By the definition of $O'$,

$$\|Yv\|_2 \leq O\left(\sup_{u \in O'} |u^\mathsf{T} Y v|\right). \tag{20}$$

The lemma is proved by combining (18), (19) and (20) and noting that $\|Y\| = \|X\|$ and $\log(1/\delta') = O(r + \log(1/\delta))$. $\qquad\square$

**Lemma C.5.** *Let $Z_1, \ldots, Z_n \in \mathbb{R}^{d_1 \times d_2}$ be mean-zero independent random matrices. Suppose for real numbers $R, p_1, \ldots, p_n \in \mathbb{R}_{\geq 0}$, we have $\Pr[\|Z_i\| \geq R] \leq p_i$. Moreover,*

$$\max\{\|\mathbb{E}[Z_i Z_i^\mathsf{T}]\|, \|\mathbb{E}[Z_i^\mathsf{T} Z_i]\|\} \leq \sigma_i^2.$$

*Then with probability at least $1 - 2(d_1 + d_2)e^{-t} - \sum_{i=1}^n p_i$,*

$$\left\|\sum_{i=1}^n Z_i\right\| \leq O\left(\sqrt{\sum_{i=1}^n \sigma_i^2 t} + Rt\right).$$

*Proof.* Since probabilities are always nonnegative, the lemma is trivial if $\sum_{i=1}^n p_i \geq 1$ or if $t \leq c$ for a sufficiently small positive constant $c$. We thus assume $\sum_{i=1}^n p_i \leq 1$ and $t \geq \Omega(1)$.

Define $Z_i' = Z_i \mathbf{1}(\|Z_i\| \leq R)$. For every unit vector $u \in \mathbb{R}^d$, by Cauchy-Schwarz,

$$\mathbb{E}[\|Z_i u\|_2 \mathbf{1}(\|Z_i\| \geq R)]^2 \leq \mathbb{E}[\|Z_i u\|_2^2] \mathbb{E}[\mathbf{1}(\|Z_i\| \geq R)^2] \leq \sigma_i^2 \Pr[\|Z_i\| \geq R] \leq \sigma_i^2 p_i.$$

By Jenson's inequality,

$$\mathbb{E}[\|Z_i u\|_2 \mathbf{1}(\|Z_i\| \geq R)] \geq \|\mathbb{E}[Z_i u \mathbf{1}(\|Z_i\| \geq R)]\|_2 = \|\mathbb{E}[Z_i \mathbf{1}(\|Z_i\| \geq R)]u\|_2.$$

Combining,

$$\|\mathbb{E}[Z_i \mathbf{1}(\|Z_i\| \geq R)]u\|_2 \leq \sigma_i \sqrt{p_i}, \quad \text{for every unit vector } u \in \mathbb{R}^{d_2},$$

which implies that $\|\mathbb{E}[Z_i']\| = \|\mathbb{E}[Z_i] - \mathbb{E}[Z_i']\| = \|\mathbb{E}[Z_i \mathbf{1}(\|Z_i\| > R)]\| \leq \sigma_i \sqrt{p_i}$.

We apply the matrix Bernstein inequality [see Vershynin, 2018, Exercise 5.4.15] to $Z_i'$. For every $i$, $\|Z_i' - \mathbb{E}[Z_i']\| \leq \|Z_i'\| + \|\mathbb{E}[Z_i']\| \leq \|Z_i'\| + \mathbb{E}[\|Z_i'\|] \leq 2R$, and

$$\mathbb{E}[(Z_i' - \mathbb{E}[Z_i'])(Z_i' - \mathbb{E}[Z_i'])^\mathsf{T}] = \mathbb{E}[Z_i'(Z_i')^\mathsf{T}] - \mathbb{E}[Z_i']\mathbb{E}[Z_i']^\mathsf{T} \preceq \mathbb{E}[Z_i'(Z_i')^\mathsf{T}] \preceq \sigma_i^2 I.$$

The matrix Bernstein inequality implies that with probability at least $1 - 2(d_1 + d_2)e^{-t}$,

$$\left\|\sum_{i=1}^n Z_i' - \sum_{i=1}^n \mathbb{E}[Z_i']\right\| \leq O\left(\sqrt{\sum_{i=1}^n \sigma_i^2 t} + Rt\right).$$

By the union bound, with probability at least $1 - 2(d_1 + d_2)e^{-t} - \sum_{i=1}^{n} p_i$,

$$\left\| \sum_{i=1}^{n} Z_i - \sum_{i=1}^{n} \mathbb{E}[Z_i'] \right\| \leq O\left( \sqrt{\sum_{i=1}^{n} \sigma_i^2 t} + Rt \right),$$

in which case,

$$\left\| \sum_{i=1}^{n} Z_i \right\| \leq \left\| \sum_{i=1}^{n} \mathbb{E}[Z_i'] \right\| + O\left( \sqrt{\sum_{i=1}^{n} \sigma_i^2 t} + Rt \right)$$

$$\leq \sum_{i=1}^{n} \sigma_i \sqrt{p_i} + O\left( \sqrt{\sum_{i=1}^{n} \sigma_i^2 t} + Rt \right)$$

$$\leq \sqrt{\left( \sum_{i=1}^{n} \sigma_i^2 \right) \left( \sum_{i=1}^{n} p_i \right)} + O\left( \sqrt{\sum_{i=1}^{n} \sigma_i^2 t} + Rt \right). \qquad \text{(by Cauchy-Schwarz)}$$

The lemma is proved by our assumption that $\sum_{i=1}^{n} p_i \leq 1$ and $t \geq \Omega(1)$. $\qquad \square$

## D   Fano's Inequality and Local Packing Numbers

For two probability distributions $p_1, p_2$ over the same set $\mathcal{X}$ with probability densities $p_1(x)$ and $p_2(x)$, their KL divergence is defined to be

$$D_{\mathrm{kl}}(p_1 \| p_2) = \mathbb{E}_{X \sim p_1} \left[ \log \frac{p_1(X)}{p_2(X)} \right].$$

For a joint distribution $p$ over random variables $X$ and $Y$, we define the mutual information between $X$ and $Y$ to be

$$I(X; Y) = D_{\mathrm{kl}}(p \| p_X \times p_Y),$$

where $p_X$ (resp. $p_Y$) is the marginal distribution of $X$ (resp. $Y$), and $p_X \times p_Y$ is the product distribution of $p_X$ and $p_Y$. The following claim is standard:

**Claim D.1.** *For random variables $X, Y, Z$, if $Y$ and $Z$ are conditionally independent given $X$, then*

$$I(X; Y, Z) \leq I(X; Y) + I(X; Z), \tag{21}$$

$$I(Y; X, Z) = I(Y; X). \tag{22}$$

**Theorem D.2** (Fano's Inequality). *Let $X, Y, \hat{X}$ be random objects. Assume their joint distribution forms a Markov chain $X \to Y \to \hat{X}$. Assume that the marginal distribution of $X$ is uniform over a finite set $\mathcal{X}$. Then,*

$$\Pr[\hat{X} \neq X] \geq 1 - \frac{I(X; Y) + \log 2}{\log |\mathcal{X}|}.$$

**Lemma D.3.** *In the setting of Theorem D.2, let $P_x$ denote the conditional distribution of $Y$ given $X = x$. Then,*

$$I(X; Y) \leq \max_{x, x' \in \mathcal{X}} D_{\mathrm{kl}}(P_{x'} \| P_x).$$

*Proof.* Let $P_x(y)$ denote $\Pr[Y = y | X = x]$. Since $X$ distributes uniformly over $\mathcal{X}$, the marginal distribution of $Y$ is given by $\Pr[Y = y] = \sum_{x \in \mathcal{X}} P_x(y)/|\mathcal{X}|$. By the definition of $I(X; Y)$,

$$I(X; Y) = \mathbb{E}_X \mathbb{E}_{Y \sim P_X} \left[ \log \frac{P_X(Y)/|\mathcal{X}|}{\sum_{x \in \mathcal{X}} P_x(Y)/|\mathcal{X}|^2} \right]$$

$$= \mathbb{E}_X \mathbb{E}_{Y \sim P_X} \left[ \log \frac{P_X(Y)}{\sum_{x \in \mathcal{X}} P_x(Y)/|\mathcal{X}|} \right]$$

$$\leq \mathbb{E}_X \mathbb{E}_{Y \sim P_X} \left[ \frac{1}{|\mathcal{X}|} \sum_{x \in \mathcal{X}} \log \frac{P_X(Y)}{P_x(Y)} \right] \qquad \text{(by Jensen's Inequality)}$$

$$= \frac{1}{|\mathcal{X}|} \sum_{x' \in \mathcal{X}} \frac{1}{|\mathcal{X}|} \sum_{x \in \mathcal{X}} \mathbb{E}_{Y \sim P_{x'}} \left[ \log \frac{P_{x'}(Y)}{P_x(Y)} \right]$$

$$= \frac{1}{|\mathcal{X}|^2} \sum_{x \in \mathcal{X}} \sum_{x' \in \mathcal{X}} D_{\mathrm{kl}}(P_{x'} \| P_x)$$

$$\leq \max_{x,x' \in \mathcal{X}} D_{\mathrm{kl}}(P_{x'} \| P_x). \qquad \square$$

**Lemma D.4** ([see Cai et al., 2013, Lemma 1]). *There exists an absolute constant $C > 0$ with the following property. For any positive integers $k \leq d$ and real number $t \in (0, 1/C]$, there exists a subset $O' \subseteq \mathbb{O}_{d,k}$ with size at least $10^{k(d-k)}$ such that*

1. *For distinct $U, V \in O'$, $\|UU^\mathsf{T} - VV^\mathsf{T}\|_F > t$.*

2. *For any $U, V \in O'$, $\|UU^\mathsf{T} - VV^\mathsf{T}\|_F \leq Ct$.*

# E   Proof of Theorem 2.1

**Theorem 2.1** (Variant of Davis–Kahan $\sin \theta$ theorem). *Let $A, \hat{A} \in \mathbb{R}^{d \times d}$ be symmetric matrices. Let $\lambda_i$ denote the $i$-th largest eigenvalue of $A$. For a positive integer $k$ smaller than $d$, let $\theta$ denote the maximum principal angle between the subspaces spanned by the top-$k$ eigenvectors of $A$ and $\hat{A}$. Assuming $\lambda_k > \lambda_{k+1}$,*

$$\sin \theta \leq \frac{2\|A - \hat{A}\|}{\lambda_k - \lambda_{k+1}}.$$

*Proof.* We assume $\frac{2\|A-\hat{A}\|}{\lambda_k - \lambda_{k+1}} < 1$ because otherwise the theorem is trivial.

Let $\hat{\lambda}_i$ denote the $i$-th largest eigenvalue of $\hat{A}$. By Weyl's inequality,

$$\hat{\lambda}_{k+1} - \lambda_{k+1} \leq \|A - \hat{A}\| \leq \frac{\lambda_k - \lambda_{k+1}}{2},$$

where we used our assumption $\frac{2\|A-\hat{A}\|}{\lambda_k - \lambda_{k+1}} < 1$ in the last inequality. This implies

$$\lambda_k - \hat{\lambda}_{k+1} \geq \frac{\lambda_k - \lambda_{k+1}}{2} > 0. \tag{23}$$

By the Davis-Kahan $\sin \theta$ theorem [Davis and Kahan, 1970],

$$\sin \theta \leq \frac{\|A - \hat{A}\|}{\lambda_k - \hat{\lambda}_{k+1}}. \tag{24}$$

Combining (23) and (24) completes the proof. $\square$

# F   Proof of Theorem 3.1

We recall:

**Theorem 3.1.** *Define $\xi^2 = \|\sum_{i=1}^n w_i^2 \mu_i \mu_i^\mathsf{T} \eta_i^2 / m_i\|$ and let $\theta$ denote the maximum principal angle between our estimator $\hat{\Gamma}$ and the true subspace $\Gamma$ spanned by $\mu_1, \ldots, \mu_n$. For any $\delta \in (0, 1/2)$, with probability at least $1 - \delta$,*

$$\sin \theta = O \left( \sigma_k^{-2} \sqrt{(d + \log(1/\delta)) \left( \xi^2 + \sum_{i=1}^n \frac{w_i^2 \eta_i^4}{m_i^2} \right)} + \sigma_k^{-2}(d + \log(1/\delta)) \max_i \frac{w_i \eta_i^2}{m_i} \right). \tag{6}$$

By (8), it suffices to upper bound the spectral norm of $A - \sum_{i=1}^n w_i \mu_i \mu_i^\mathsf{T}$. We achieve this goal by proving Lemmas F.1 to F.3 in which we bound the spectral norm of each term on the right-hand-side of the following decomposition:

$$A - \sum_{i=1}^n w_i \mu_i \mu_i^\mathsf{T} = \sum_{i=1}^n w_i \mu_i \bar{z}_i^\mathsf{T} + \sum_{i=1}^n w_i \bar{z}_i \mu_i^\mathsf{T} + \sum_{i=1}^n \frac{w_i}{m_i(m_i - 1)} \sum_{j_1 \neq j_2} z_{ij_1} z_{ij_2}, \tag{25}$$

where $\bar{z}_i = \frac{1}{m_i} \sum_{j=1}^{m_i} z_{ij}$.

**Lemma F.1.** *In the setting of Theorem 3.1, for any $\delta \in (0, 1/2)$, with probability at least $1 - \delta$,*

$$\| \sum_{i=1}^n w_i \mu_i \bar{z}_i^\mathsf{T} \| \leq O\left( \xi \sqrt{d + \log(1/\delta)} \right).$$

*Proof.* Let $O'$ denote the set $O_d' \subseteq \mathbb{O}_{d,1}$ guaranteed by Lemma C.1. For any fixed $u, v \in O'$, by Lemma B.1, $\bar{z}_i^\mathsf{T} v$ is $\eta_i/\sqrt{m_i}$-sub-Gaussian, and thus $w_i(\mu_i^\mathsf{T} u)(\bar{z}_i^\mathsf{T} v)$ is $w_i|\mu_i^\mathsf{T} u|\eta_i/\sqrt{m_i}$-sub-Gaussian. By Lemma B.3, for any $\delta' \in (0, 1/2)$, with probability at least $1 - \delta'$,

$$\left| \sum_{i=1}^n w_i(\mu_i^\mathsf{T} u)(\bar{z}_i^\mathsf{T} v) \right| \leq O\left( \sqrt{\log(1/\delta') \sum_{i=1}^n w_i^2(\mu_i^\mathsf{T} u)^2 \eta_i^2/m_i} \right). \tag{26}$$

By the definition of $\xi$,

$$\sum_{i=1}^n w_i^2(\mu_i^\mathsf{T} u)^2 \eta_i^2/m_i = u^\mathsf{T} \left( \sum_{i=1}^n w_i^2 \mu_i \mu_i^\mathsf{T} \eta_i^2/m_i \right) u \leq \left\| \sum_{i=1}^n w_i^2 \mu_i \mu_i^\mathsf{T} \eta_i^2/m_i \right\| = \xi^2.$$

Plugging this into (26), and setting $\delta' = \delta/|O'|^2$, with probability at least $1 - \delta/|O'|^2$,

$$\left| \sum_{i=1}^n w_i(\mu_i^\mathsf{T} u)(\bar{z}_i^\mathsf{T} v) \right| \leq O\left( \xi \sqrt{\log(1/\delta')} \right) \leq O\left( \xi \sqrt{d + \log(1/\delta)} \right),$$

where we used the fact that $\log |O'| = O(d)$ to obtain the last inequality. By a union bound over $u, v \in O'$, with probability at least $1 - \delta$, $\sup_{u \in O'} \sup_{v \in O'} |u^\mathsf{T} \left( \sum_{i=1}^n w_i \mu_i \bar{z}_i^\mathsf{T} \right) v| \leq O(\xi \sqrt{d + \log(1/\delta)})$. The lemma is proved by combining this with $\| \sum_{i=1}^n w_i \mu_i \bar{z}_i^\mathsf{T} \| \leq \sup_{u \in O'} \sup_{v \in O'} |u^\mathsf{T} \left( \sum_{i=1}^n w_i \mu_i \bar{z}_i^\mathsf{T} \right) v|$ by Lemma C.1. $\qquad\square$

**Lemma F.2.** *In the setting of Theorem 3.1, define $Z = \sum_{i=1}^n \frac{w_i}{m_i(m_i-1)} (\sum_{j=1}^{m_i} z_{ij})(\sum_{j=1}^{m_i} z_{ij})^\mathsf{T}$. For any $\delta \in (0, 1/2)$, with probability at least $1 - \delta$,*

$$\|Z - \mathbb{E}[Z]\|$$
$$\leq O\left( \sqrt{(d + \log(1/\delta)) \sum_{i=1}^n w_i^2 \eta_i^4/m_i^2} + (d + \log(1/\delta)) \max_i w_i \eta_i^2/m_i \right). \tag{27}$$

*Proof.* By Lemma B.1, $\sum_{j=1}^{m_i} z_{ij}$ has sub-Gaussian constant $b_i$ for some $b_i \in \mathbb{R}_{\geq 0}$ satisfying $b_i = O(\sqrt{m_i}\eta_i)$. Define $\hat{z}_i = \frac{1}{b_i} \sum_{j=1}^{m_i} z_{ij}$ and now $\hat{z}_i$ has sub-Gaussian constant 1. We write $Z$ in terms of $\hat{z}_i$:

$$Z = \sum_{i=1}^n \frac{w_i b_i^2}{m_i(m_i-1)} \hat{z}_i \hat{z}_i^\mathsf{T}.$$

Applying Lemma C.2 completes the proof. $\qquad\square$

**Lemma F.3.** *In the setting of Theorem 3.1, define $Z = \sum_{i=1}^n \frac{w_i}{m_i(m_i-1)} \sum_{j=1}^{m_i} z_{ij} z_{ij}^\mathsf{T}$. For any $\delta \in (0, 1/2)$, with probability at least $1 - \delta$,*

$$\|Z - \mathbb{E}[Z]\|$$
$$\leq O\left( \sqrt{(d + \log(1/\delta)) \sum_{i=1}^n w_i^2 \eta_i^4/m_i^3} + (d + \log(1/\delta)) \max_i w_i \eta_i^2/m_i^2 \right). \tag{28}$$

*Proof.* Define $\hat{z}_{ij} = \frac{1}{\eta_i} z_{ij}$ and now $\hat{z}_{ij}$ has sub-Gaussian constant 1. We write $Z$ in terms of $\hat{z}_{ij}$:

$$Z = \sum_{i=1}^n \sum_{j=1}^{m_i} \frac{w_i \eta_i^2}{m_i(m_i-1)} \hat{z}_{ij} \hat{z}_{ij}^\mathsf{T}.$$

Applying Lemma C.2 completes the proof. $\qquad\square$

We are now ready to finish the proof of Theorem 3.1.

*Proof of Theorem 3.1.* We start with the last term in (25):

$$\sum_{i=1}^{n} \frac{w_i}{m_i(m_i-1)} \sum_{j_1 \neq j_2} z_{ij_1} z_{ij_2}$$

$$= \sum_{i=1}^{n} \frac{w_i}{m_i(m_i-1)} \left(\sum_{j=1}^{m_i} z_{ij}\right)\left(\sum_{j=1}^{m_i} z_{ij}\right)^{\mathsf{T}} - \sum_{i=1}^{n} \frac{w_i}{m_i(m_i-1)} \sum_{j=1}^{m_i} z_{ij} z_{ij}^{\mathsf{T}}.$$

It is clear that $\mathbb{E}\left[\sum_{i=1}^{n} \frac{w_i}{m_i(m_i-1)} \sum_{j_1 \neq j_2} z_{ij_1} z_{ij_2}\right] = 0$, so combining Lemmas F.2 and F.3, with probability at least $1 - \delta$,

$$\left\|\sum_{i=1}^{n} \frac{w_i}{m_i(m_i-1)} \sum_{j_1 \neq j_2} z_{ij_1} z_{ij_2}\right\| \leq O\left(\sqrt{(d+\log(1/\delta))\sum_{i=1}^{n} w_i^2 \eta_i^4/m_i^2} + (d+\log(1/\delta))\max_i w_i \eta_i^2/m_i\right),$$

(29)

where we used the fact that the right-hand-side of (27) is at most the right-hand-side of (28) (up to a constant factor).

By Lemma F.1,

$$\left\|\sum_{i=1}^{n} w_i \mu_i \bar{z}_i^{\mathsf{T}} + \sum_{i=1}^{n} w_i \bar{z}_i \mu_i^{\mathsf{T}}\right\| \leq 2\left\|\sum_{i=1}^{n} w_i \mu_i \bar{z}_i^{\mathsf{T}}\right\| \leq O\left(\xi\sqrt{d+\log(1/\delta)}\right). \qquad (30)$$

Plugging (29) and (30) into (25) and then into (8) proves the theorem. $\qquad \square$

# G  Proof of Corollary 3.2

*Proof.* By the definition of $\xi^2$ in Theorem 3.1, $\xi^2 = \|\sum_{i=1}^{n} \frac{\mu_i \mu_i^{\mathsf{T}} t^2}{n^2}\| \leq \frac{t^2 \sigma_1^2}{n}$. Plugging this into (6), we know that the following inequality holds with probability at least $1 - \delta$:

$$\sin\theta \leq O\left(\sigma_k^{-2}\sqrt{(d+\log(1/\delta))\left(\frac{t^2 \sigma_1^2}{n} + \frac{t^4}{n}\right)} + \sigma_k^{-2}(d+\log(1/\delta))\frac{t^2}{n}\right)$$

$$= O\left(\frac{t\sigma_1 + t^2}{\sigma_k^2}\sqrt{\frac{d+\log(1/\delta)}{n}} + \sigma_k^{-2}(d+\log(1/\delta))\frac{t^2}{n}\right). \qquad (31)$$

Since $\sin\theta \leq 1$ always holds, inequality (31) implies

$$\sin\theta \leq O\left(\frac{t\sigma_1 + t^2}{\sigma_k^2}\sqrt{\frac{d+\log(1/\delta)}{n}} + \min\left\{1, \sigma_k^{-2}(d+\log(1/\delta))\frac{t^2}{n}\right\}\right) \qquad (32)$$

Since $\min\{1, y\} \leq \sqrt{y}$ for every $y \in \mathbb{R}_{\geq 0}$, we have

$$\min\left\{1, \sigma_k^{-2}(d+\log(1/\delta))\frac{t^2}{n}\right\} \leq \frac{t}{\sigma_k}\sqrt{\frac{(d+\log(1/\delta))}{n}} \leq \frac{t\sigma_1}{\sigma_k^2}\sqrt{\frac{(d+\log(1/\delta))}{n}}. \qquad (33)$$

Plugging (33) into (32) proves the corollary. $\qquad \square$

# H  Proof of Claim 3.3

*Proof.* Without loss of generality, we can assume that the subspace $\Gamma$ in Assumption 3.1 is the subspace containing all vectors with all but the first $k$ coordinates being zeros. We let $\hat{\mu}_i \in \mathbb{R}^k$ denote the first $k$ coordinates of $\mu_i$. By Lemma C.2, with probability at least $1 - \delta/2$,

$$\left\|\sum_{i=1}^{n} w_i \hat{\mu}_i \hat{\mu}_i^{\mathsf{T}} - \mathbb{E}\left[\sum_{i=1}^{n} w_i \hat{\mu}_i \hat{\mu}_i^{\mathsf{T}}\right]\right\|$$

$$\leq O\left(\sigma^2\left(\sqrt{(k+\log(1/\delta))\sum_{i=1}^n w_i^2} + (k+\log(1/\delta))\max_i |w_i|\right)\right)$$

$$\leq O\left(\sigma^2\left(\sqrt{\frac{1}{C_*}} + \frac{1}{C_*}\right)\right), \tag{34}$$

where we used the fact that $\sum_{i=1}^n w_i^2 \leq \max_i w_i \leq 1/C^*(k+\log(1/\delta))$. Choosing $C_*$ large enough, (34) implies

$$\left\|\sum_{i=1}^n w_i\hat{\mu}_i\hat{\mu}_i^\mathsf{T} - \mathbb{E}\left[\sum_{i=1}^n w_i\hat{\mu}_i\hat{\mu}_i^\mathsf{T}\right]\right\| \leq \sigma^2/2. \tag{35}$$

By Assumption 3.1, the $k$-th largest eigenvalue of $\mathbb{E}[\mu_i\mu_i^\mathsf{T}]$ is at least $\sigma^2$. This implies that the $k$-th largest eigenvalue of $\mathbb{E}[\sum_{i=1}^n w_i\hat{\mu}_i\hat{\mu}_i^\mathsf{T}]$ is at least $\sigma^2$. The claim is proved by combining this with (35) and noting that $\sum_{i=1}^n w_i\hat{\mu}_i\hat{\mu}_i^\mathsf{T}$ and $\sum_{i=1}^n w_i\mu_i\mu_i^\mathsf{T}$ have the same $k$-th largest eigenvalue. $\qquad\square$

# I  Proof of Theorem 3.4

**Theorem 3.4.** *Under Assumptions 3.1 and 3.2, if we choose $w_1, \ldots, w_n$ as in (9) and define $\theta = \angle(\Gamma, \hat{\Gamma})$, for $\delta \in (0, 1/2)$, with probability at least $1 - \delta$,*

$$\sin\theta \leq O\left(\sqrt{\frac{d + \log(1/\delta)}{\sum_{i=1}^n \gamma_i'}}\right). \tag{10}$$

*Proof.* By Assumption 3.2 and our choice of $w_i$, we have $w_i \leq 1/C_*(k + \log(1/\delta))$. By the definition of $C_*$, with probability at least $1 - \delta/2$,

$$\sigma_k^2 \geq \sigma^2/2. \tag{36}$$

By Lemma C.3, for $\xi^2$ defined in Theorem 3.1, with probability at least $1 - \delta/4$,

$$\xi^2 \leq O\left(\sum_{i=1}^n \frac{w_i^2\eta_i^2\sigma^2}{m_i} + (k+\log(1/\delta))\max_i \frac{w_i^2\eta_i^2\sigma^2}{m_i}\right). \tag{37}$$

Replacing $\delta$ in Theorem 3.1 by $\delta/4$ and plugging (36) and (37) into (6), by the union bound, with probability at least $1 - \delta$,

$$\sin\theta \leq O\Bigg(\sqrt{(d+\log(1/\delta))\sum_{i=1}^n\left(\frac{w_i^2\eta_i^2}{\sigma^2 m_i} + \frac{w_i^2\eta_i^4}{\sigma^4 m_i^2}\right)}$$

$$+ \sqrt{(d+\log(1/\delta))(k+\log(1/\delta))\max_i \frac{w_i^2\eta_i^2}{\sigma^2 m_i}} + (d+\log(1/\delta))\max_i \frac{w_i\eta_i^2}{\sigma^2 m_i}\cdot\Bigg) \tag{38}$$

By definition, $\frac{\eta_i^2}{\sigma^2 m_i} + \frac{\eta_i^4}{\sigma^4 m_i^2} = 1/\gamma_i \leq 1/\gamma_i'$. Therefore, inequality (38) implies

$$\sin\theta$$

$$\leq O\Bigg(\sqrt{(d+\log(1/\delta))\sum_{i=1}^n \frac{w_i^2}{\gamma_i'}} + \sqrt{(d+\log(1/\delta))(k+\log(1/\delta))\max_i \frac{w_i^2}{\gamma_i'}} + (d+\log(1/\delta))\max_i \frac{w_i}{\gamma_i'}\Bigg).$$

Plugging $w_i = \frac{\gamma_i'}{\sum_{\ell=1}^n \gamma_\ell'}$ into the inequality above,

$$\sin\theta \leq O\left(\sqrt{\frac{d+\log(1/\delta)}{\sum_{i=1}^n \gamma_i'}} + \sqrt{\frac{(d+\log(1/\delta))(k+\log(1/\delta))\max_i \gamma_i'}{\left(\sum_{i=1}^n \gamma_i'\right)^2}} + \frac{d+\log(1/\delta)}{\sum_{i=1}^n \gamma_i'}\right).$$

$$\tag{39}$$

Since $\sin\theta \le 1$ always holds, inequality (39) implies

$$\sin\theta \le O\left(\sqrt{\frac{d+\log(1/\delta)}{\sum_{i=1}^{n}\gamma_i'}} + \sqrt{\frac{(d+\log(1/\delta))(k+\log(1/\delta))\max_i \gamma_i'}{\left(\sum_{i=1}^{n}\gamma_i'\right)^2}} + \min\left\{1, \frac{d+\log(1/\delta)}{\sum_{i=1}^{n}\gamma_i'}\right\}\right).$$

The theorem is proved by the following facts:

$$(k+\log(1/\delta))\max_i \gamma_i' \le O\left(\sum_{i=1}^{n}\gamma_i'\right), \qquad\qquad \text{(by Assumption 3.2)}$$

$$\min\left\{1, \frac{d+\log(1/\delta)}{\sum_{i=1}^{n}\gamma_i'}\right\} \le \sqrt{\frac{d+\log(1/\delta)}{\sum_{i=1}^{n}\gamma_i'}}. \qquad\qquad \square$$

## J  Proof of Lemma 3.8

*Proof.* The KL divergence between mean-zero Gaussians can be computed by the following formula:

$$D_{\mathrm{kl}}(N(0,\hat{\Sigma})\|N(0,\Sigma)) = \frac{1}{2}\left(\mathrm{tr}(\Sigma^{-1}\hat{\Sigma}) - d + \log\frac{\det\Sigma}{\det\hat{\Sigma}}\right). \tag{40}$$

It is clear that

$$\det\Sigma = \det\hat{\Sigma} = (\sigma^2 + \eta^2)^k (\eta^2)^{d-k}. \tag{41}$$

It remains to compute $\mathrm{tr}(\Sigma^{-1}\hat{\Sigma}) - d$.

Define $J = \begin{bmatrix} I_k & 0 \\ 0 & 0 \end{bmatrix} \in \mathbb{R}^{d\times d}$. By the definition of principal angles in Section 2, there exists $P, \hat{P} \in \mathbb{O}_d$ such that $UU^{\mathsf{T}} = PJP^{\mathsf{T}}, \hat{U}\hat{U}^{\mathsf{T}} = \hat{P}J\hat{P}^{\mathsf{T}}$ and

$$P^{\mathsf{T}}\hat{P} = \begin{bmatrix} I_{k-\ell} & 0 & 0 & 0 \\ 0 & \cos\Theta & \sin\Theta & 0 \\ 0 & -\sin\Theta & \cos\Theta & 0 \\ 0 & 0 & 0 & I_{d-\hat{k}-\ell} \end{bmatrix},$$

where $\theta_1,\ldots,\theta_\ell$ are the principal angles between $\mathrm{col}(U)$ and $\mathrm{col}(\hat{U})$, $\cos\Theta = \mathrm{diag}(\cos\theta_1,\ldots,\cos\theta_\ell)$, and $\sin\Theta = \mathrm{diag}(\sin\theta_1,\ldots,\sin\theta_\ell)$. Now we have $\hat{U}\hat{U}^{\mathsf{T}} = \hat{P}J\hat{P}^{\mathsf{T}} = P(P^{\mathsf{T}}\hat{P})J(P^{\mathsf{T}}\hat{P})^{\mathsf{T}}P^{\mathsf{T}} = P\hat{J}P^{\mathsf{T}}$ where

$$\hat{J} = (P^{\mathsf{T}}\hat{P})J(P^{\mathsf{T}}\hat{P})^{\mathsf{T}} = \begin{bmatrix} I_{k-\ell} & 0 & 0 & 0 \\ 0 & (\cos\Theta)^2 & -\cos\Theta\sin\Theta & 0 \\ 0 & -\cos\Theta\sin\Theta & (\sin\Theta)^2 & 0 \\ 0 & 0 & 0 & 0 \end{bmatrix}.$$

Therefore, $\Sigma = P(\sigma^2 J + \eta^2 I)P^{\mathsf{T}}$ and $\hat{\Sigma} = P(\sigma^2\hat{J} + \eta^2 I)P^{\mathsf{T}}$, which implies

$$\Sigma^{-1}\hat{\Sigma} = P(\sigma^2 J + \eta^2 I)^{-1}(\sigma^2\hat{J} + \eta^2 I)P^{\mathsf{T}}.$$

Since $\sigma^2 J + \eta^2 I = \mathrm{diag}(\underbrace{\sigma^2 + \eta^2,\ldots,\sigma^2+\eta^2}_{k}, \underbrace{\eta^2,\ldots,\eta^2}_{d-k})$ and the diagonal entries of $\sigma^2\hat{J} + \eta^2 I$ are

$$\underbrace{\sigma^2+\eta^2,\ldots,\sigma^2+\eta^2}_{k-\ell}, \underbrace{\sigma^2\cos^2\theta_1+\eta^2,\ldots,\sigma^2\cos^2\theta_\ell+\eta^2}_{\ell},$$

$$\underbrace{\sigma^2\sin^2\theta_1+\eta^2,\ldots,\sigma^2\sin^2\theta_\ell+\eta^2}_{\ell}, \underbrace{\eta^2,\ldots,\eta^2}_{d-k-\ell},$$

we have

$$\mathrm{tr}(\Sigma^{-1}\hat{\Sigma}) - d = \mathrm{tr}((\sigma^2 J + \eta^2 I)^{-1}(\sigma^2\hat{J} + \eta^2 I)) - d$$

$$= \sum_{i=1}^{\ell} \left( \frac{\sigma^2 \cos^2 \theta_i + \eta^2}{\sigma^2 + \eta^2} - 1 \right) + \sum_{i=1}^{\ell} \left( \frac{\sigma^2 \sin^2 \theta_i + \eta^2}{\eta^2} - 1 \right)$$

$$= \sum_{i=1}^{\ell} \sin^2 \theta_i \left( \frac{\sigma^2}{\eta^2} - \frac{\sigma^2}{\sigma^2 + \eta^2} \right)$$

$$= \frac{\sigma^4 \|UU^\mathsf{T} - \hat{U}\hat{U}^\mathsf{T}\|_F^2}{2(\eta^2 \sigma^2 + \eta^4)}. \tag{42}$$

Plugging (41) and (42) into (40) completes the proof. □

## K   Proof of Theorem 3.7

We recall Theorem 3.7:

**Theorem 3.7.** *Let $k, d, n$ be positive integers satisfying $k < d$ and $k \leq n$. Let $m_1, \ldots, m_n$ be positive integers and $\sigma, \eta_1, \ldots, \eta_n$ be positive real numbers. Suppose we draw $U \in \mathbb{O}_{d,k}$ from $\mathsf{Haar}(\mathbb{O}_{d,k})$ and then draw $\mu_1, \ldots, \mu_n$ independently from $N(0, \sigma^2 UU^\mathsf{T})$. For every $i = 1, \ldots, n$, we draw $m_i$ data points $x_{ij}$ for $j = 1, \ldots, m_i$ as $x_{ij} = \mu_i + z_{ij}$, where each $z_{ij}$ is drawn independently from the spherical Gaussian $N(0, \eta_i^2 I)$. Let $\hat{\Gamma}$ be any estimator mapping $(x_{ij})_{1 \leq i \leq n, 1 \leq j \leq m_i}$ to a (possibly randomized) $k$-dimensional subspace of $\mathbb{R}^d$. Let $\theta$ denote the maximum principal angle between $\hat{\Gamma}((x_{ij})_{1 \leq i \leq n, 1 \leq j \leq m_i})$ and the true subspace $\Gamma = \mathsf{col}(U)$. If real numbers $t \geq 0$ and $\delta \in [0, 1/2)$ satisfy $\Pr[\sin \theta \leq t] \geq 1 - \delta$, then*

$$t \geq \Omega \left( \min \left\{ 1, \sqrt{\frac{(d-k)(1-\delta)}{\sum_{i=k}^n \gamma_i}} \right\} \right), \tag{11}$$

*where $\gamma_1, \ldots, \gamma_n$ are defined in Definition 3.1.*

*Proof.* The theorem is trivial if $t$ is lower bounded by a positive absolute constant, so without loss of generality, we can assume that $t \leq 1/(2\sqrt{2}C)$ for the constant $C$ in Lemma D.4.

Now we describe the distribution of $E$ on which we prove the error lower bound (11) that does not depend on $\gamma_1, \ldots, \gamma_{k-1}$. As we mentioned in Section 3.3, the first $k-1$ columns of $E$ are fixed to be $\begin{bmatrix} I_{k-1} \\ 0 \end{bmatrix}$, so we only need to describe the distribution of the $k$-th column of $E$. By Lemma D.4, there exists $O' \subseteq \mathbb{O}_{d-k+1,1}$ with size at least $10^{d-k}$ such that

1. for distinct $u, v \in O'$, $\|uu^\mathsf{T} - vv^\mathsf{T}\|_F > 2\sqrt{2}t$;

2. for any $u, v \in O'$, $\|uu^\mathsf{T} - vv^\mathsf{T}\|_F \leq O(t)$.

The $k$-th column of $E$ is chosen by first drawing a uniform random $u \in O'$ and then prepend zeros to it. That is

$$E = \begin{bmatrix} I_{k-1} & 0 \\ 0 & u \end{bmatrix}. \tag{43}$$

We let $\mathcal{E}$ denote the support of the $E$ so that $E$ is chosen uniformly at random from $\mathcal{E}$.

By the law of total expectation, conditioned on $W, V_1, (\mu_1, \ldots, \mu_{k-1})$ and $(x_{ij})_{1 \leq i < k, 1 \leq j \leq m_i}$ being fixed to some specific value, we still have $\Pr[\sin \theta \geq t] \leq \delta$, where now $\Pr[\cdot]$ represents the conditional probability. We will abuse notation and omit explicitly writing out the conditioning throughout the proof.

After the conditioning, the randomness in $V$ comes only from the randomness in $E \in \mathcal{E}$, and the distribution of $V$ is the uniform distribution over the set $\mathcal{V} := \{WE : E \in \mathcal{E}\}$. For any two matrices $V', V'' \in \mathcal{V}$, there exists $E', E'' \in \mathcal{E}$ such that $V' = WE'$ and $V'' = WE''$. Moreover, there exists $u, v \in O'$ such that $E' = \begin{bmatrix} I_{k-1} & 0 \\ 0 & u \end{bmatrix}$ and $E'' = \begin{bmatrix} I_{k-1} & 0 \\ 0 & v \end{bmatrix}$. This implies that

$$\|V'(V')^\mathsf{T} - V''(V'')^\mathsf{T}\|_F = \|E'(E')^\mathsf{T} - E''(E'')^\mathsf{T}\|_F = \|uu^\mathsf{T} - vv^\mathsf{T}\|_F \leq O(t). \tag{44}$$

Moreover, when $V' \neq V''$, we have $u \neq v$ and thus

$$\|V'(V')^\mathsf{T} - V''(V'')^\mathsf{T}\| = \|E'(E')^\mathsf{T} - E''(E'')^\mathsf{T}\| = \|uu^\mathsf{T} - vv^\mathsf{T}\| > 2t, \tag{45}$$

where we used the fact that $\|uu^\mathsf{T} - vv^\mathsf{T}\| = \sin\angle(u,v)$ and $2\sqrt{2}t < \|uu^\mathsf{T} - vv^\mathsf{T}\|_F = \sqrt{2\sin^2\angle(u,v)}$. We define $\hat{V} \in \mathcal{V}$ so that the maximum principal angle between $\mathsf{col}(\hat{V})$ and $\hat{\Gamma}$ is minimized. When $\sin\theta \leq t$ (i.e., $\sin\angle(\mathsf{col}(V),\Gamma) \leq t$), for any $V' \in \mathcal{V}$ different from $V$,

$$\sin\angle(\mathsf{col}(V'),\Gamma) \geq \sin\angle(V,V') - \sin\angle(\mathsf{col}(V),\Gamma) > 2t - t = t \geq \sin\angle(\mathsf{col}(V),\Gamma),$$

where we used $\sin\angle(V,V') = \|VV^\mathsf{T} - V'(V')^\mathsf{T}\| > 2t$ by (45). Therefore, $\sin\theta \leq t$ implies $\hat{V} = V$ and thus $\Pr[\sin\theta > t] \geq \Pr[\hat{V} \neq V]$.

Now we apply Theorem D.2 to the following Markov chain

$$V \rightarrow (x_{ij})_{k \leq i \leq n, 1 \leq j \leq m_i} \rightarrow \hat{V}$$

to get

$$\delta \geq \Pr[\sin\theta > t] \geq \Pr[\hat{V} \neq V] \geq 1 - \frac{I(V;(x_{ij})_{k \leq i \leq n, 1 \leq j \leq m_i}) + \log 2}{\log|\mathcal{V}|}. \tag{46}$$

Now we bound $I(V;(x_{ij})_{k \leq i \leq n, 1 \leq j \leq m_i})$. According to the graphical model in Figure 3, $\mu_k,\ldots,\mu_n$ are conditionally independent given $V$, and thus $(x_{ij})_{1 \leq j \leq m_i}$ are conditionally independent for $i = k,\ldots,n$ given $V$. By (21),

$$I(V;(x_{ij})_{k \leq i \leq n, 1 \leq j \leq m_i}) \leq \sum_{i=k}^{n} I(V;(x_{ij})_{1 \leq j \leq m_i}). \tag{47}$$

Let $\bar{x}_i$ denote $\frac{1}{m_i}\sum_{j=1}^{m_i} x_{ij}$. Since $x_{ij}$ for $j = 1,\ldots,m_i$ are drawn iid from $N(\mu_i, \eta_i^2 I)$, it is a standard fact that $\bar{x}_i$ is a sufficient statistic for $\mu_i$, i.e., $(x_{ij})_{1 \leq j \leq m_i}$ and $\mu_i$ are conditionally independent given $\bar{x}_i$. Therefore, for any measurable set $S \in (\mathbb{R}^d)^{m_i}$,

$$\Pr[(x_{ij})_{1 \leq j \leq m_i} \in S | \mu_i, \bar{x}_i] = \Pr[(x_{ij})_{1 \leq j \leq m_i} \in S | \bar{x}_i]. \tag{48}$$

Since $(x_{ij})_{j=1,\ldots,m_i}$ and $V$ are conditionally independent given $\mu_i$,

$$\Pr[(x_{ij})_{1 \leq j \leq m_i} \in S | V, \mu_i, \bar{x}_i] = \Pr[(x_{ij})_{1 \leq j \leq m_i} \in S | \mu_i, \bar{x}_i]. \tag{49}$$

Combining (48) and (49), $(x_{ij})_{1 \leq j \leq m_i}$ and $V$ are conditionally independent given $\bar{x}_i$. By (22),

$$I(V;(x_{ij})_{1 \leq j \leq m_i}) = I(V;(x_{ij})_{1 \leq j \leq m_i}, \bar{x}_i) = I(V; \bar{x}_i). \tag{50}$$

Since $\bar{x}_i = \mu_i + \frac{1}{m_i}\sum_{j=1}^{m_i} z_j$, the conditional distribution of $\bar{x}_i$ given $V$ is $N(0, \sigma^2 VV^\mathsf{T} + \frac{\eta_i^2}{m_i}I)$. By Lemma 3.8 and inequality (44),

$$I(V;\bar{x}_i) \leq \sup_{V',V'' \in \mathcal{V}} D_{\mathrm{kl}}\left(N\left(0, \sigma^2 V'(V')^\mathsf{T} + \frac{\eta_i^2}{m_i}I\right)\,\Big\|\,N\left(0, \sigma^2 V''(V'')^\mathsf{T} + \frac{\eta_i^2}{m_i}I\right)\right) \leq O(t^2 \gamma_i). \tag{51}$$

Combining (47), (50), and (51),

$$I(V;(x_{ij})_{k \leq i \leq n, 1 \leq j \leq m_i}) \leq O\left(t^2 \sum_{i=k}^{n} \gamma_i\right).$$

The theorem is proved by plugging this into (46) and noting that $\log|\mathcal{V}| \geq 2(d-k)$. $\square$

## L  Our Results in the Linear Models Setting

In the linear models setting (see Section 4), our estimator is the subspace $\hat{\Gamma}$ spanned by the top-$k$ eigenvectors of $A$ defined in (3). We use $\theta$ to denote the maximum principal angle between our estimator $\hat{\Gamma}$ and the true subspace $\Gamma$, and define $\sigma_k^2$ to be the $k$-th largest eigenvalue of $\sum_{i=1}^{n} w_i \beta_i \beta_i^\mathsf{T}$. To describe our guarantee on $\theta$, it is convenient to first define the following quantities for every $i = 1, \ldots, n$:

$$p_i = \frac{(d + \log(n/\delta))(m_i + \log(n/\delta))}{m_i^2}, \quad q_i = \sqrt{\frac{d + \log(n/\delta)}{m_i}} + p_i.$$

We prove the following guarantees for our estimator in Appendices L.1 and L.2:

**Theorem L.1.** *There exists an absolute constant $C > 0$ such that for any $\delta \in (0, 1/2)$, with probability at least $1 - \delta$,*

$$\sin \theta \leq C\sigma_k^{-2} \sqrt{\sum_{i=1}^{n} w_i^2 d \log(d/\delta) \left( \frac{\|\beta_i\|_2^4 + \eta_i^2 \|\beta_i\|_2^2}{m_i} + \frac{\eta_i^4}{m_i^2} \right)}$$
$$+ C\sigma_k^{-2} \max_i w_i \log(d/\delta)(\|\beta_i\|_2^2 q_i + \eta_i \|\beta_i\|_2 q_i + \eta_i^2 p_i).$$

**Corollary L.2.** *Suppose $m \in \mathbb{Z}_{>0}$ and $\eta \in \mathbb{R}_{>0}$ satisfy $2 \leq m \leq \min_{1 \leq i \leq n} m_i$ and $\max_{1 \leq i \leq n} \eta_i \leq \eta$. Choosing $w_1 = \cdots = w_n = 1/n$, for any $\delta \in (0, 1/2)$, with probability at least $1 - \delta$,*

$$\sin \theta \leq O \left( \log^3(nd/\delta) \left( \sqrt{\frac{d(\frac{1}{n}\sum_{i=1}^{n} \|\beta_i\|_2^4 + \frac{1}{n}\sum_{i=1}^{n} \eta^2 \|\beta_i\|_2^2 + \eta^4/m)}{mn\sigma_k^4}} + \frac{d \max_i \|\beta_i\|_2^2}{mn\sigma_k^2} \right) \right). \tag{52}$$

*If we further assume that $\max_{1 \leq i \leq n} \|\beta_i\|_2 \leq r$, then (52) implies*

$$\sin \theta \leq O \left( \log^3(nd/\delta) \left( \sqrt{\frac{d(r^4 + r^2\eta^2 + \eta^4/m)}{mn\sigma_k^4}} \right) \right). \tag{53}$$

In [Tripuraneni et al., 2021], it is further assumed that $\eta = \Theta(1), r = \Theta(1), \|\beta_i\|_2 = \Theta(1), \delta = (mn)^{-100}$. Defining $\kappa = \frac{1}{n}\sum_{i=1}^{n} \|\beta_i\|_2^2 / k\sigma_k^2 = \Theta(1/k\sigma_k^2)$ as in [Tripuraneni et al., 2021] and using the fact $\sum_{i=1}^{n} \|\beta_i\|_2^4 \leq (\max_i \|\beta_i\|_2^2) \sum_{i=1}^{n} \|\beta_i\|_2^2$, our bound becomes

$$\sin \theta \leq O \left( \sqrt{\frac{d}{mn\sigma_k^4}} \log^3(mn) \right) = O \left( \sqrt{\frac{\kappa}{\sigma_k^2} \frac{dk}{mn}} \log^3(mn) \right),$$

matching the bound in [Tripuraneni et al., 2021, Theorem 3].

As in the PCA setting, our proof of Theorem L.1 is based on the Davis-Kahan $\sin \theta$ theorem (Theorem 2.1) and a bound on the spectral norm of the difference between the matrix $A$ and its expectation. We prove and make crucial use of a generalization of the matrix Bernstein inequality (Lemma C.5) which turns out to be slightly more convenient than a similar inequality used in [Tripuraneni et al., 2021].

### L.1 Proof of Theorem L.1

Plugging $y_{ij} = x_{ij}^\mathsf{T}\beta_i + z_{ij}$ into the definition of $A$,

$$A = \sum_{i=1}^{n} \frac{w_i}{m_i(m_i - 1)} \sum_{j_1 \neq j_2} x_{ij_1}(x_{ij_1}^\mathsf{T}\beta_i + z_{ij_1})(\beta_i^\mathsf{T} x_{ij_2} + z_{ij_2})x_{ij_2}^\mathsf{T}.$$

When $j_1 \neq j_2$, conditioned on $x_{ij_1}$ and $x_{ij_2}$, the noise terms $z_{ij_1}$ and $z_{ij_2}$ are independent and have zero mean. Therefore,

$$\mathbb{E}[x_{ij_1}(x_{ij_1}^\mathsf{T}\beta_i + z_{ij_1})(\beta_i^\mathsf{T} x_{ij_2} + z_{ij_2})x_{ij_2}^\mathsf{T}] = \mathbb{E}[x_{ij_1}x_{ij_1}^\mathsf{T}\beta_i\beta_i^\mathsf{T} x_{ij_2}x_{ij_2}^\mathsf{T}] = \mathbb{E}[\beta_i\beta_i^\mathsf{T}],$$

where we used the fact that $x_{ij_1}$ and $x_{ij_2}$ are independent and have zero mean and identity covariance matrix. Thus, the expectation of $A$ is $\bar{A} = \sum_{i=1}^{n} w_i\beta_i\beta_i^\mathsf{T}$, and as in the PCA setting, our goal is to bound the spectral norm of the difference $A - \bar{A}$. We decompose $A$ as

$$A = E + F + F^\mathsf{T} + G, \text{ where} \tag{54}$$

$$E = \sum_{i=1}^{n} \frac{w_i}{m_i(m_i - 1)} \sum_{j_1 \neq j_2} x_{ij_1}x_{ij_1}^\mathsf{T}\beta_i\beta_i^\mathsf{T} x_{ij_2}x_{ij_2}^\mathsf{T},$$

$$F = \sum_{i=1}^{n} \frac{w_i}{m_i(m_i - 1)} \sum_{j_1 \neq j_2} x_{ij_1}x_{ij_1}^\mathsf{T}\beta_i z_{ij_2}x_{ij_2}^\mathsf{T},$$

$$G = \sum_{i=1}^{n} \frac{w_i}{m_i(m_i-1)} \sum_{j_1 \neq j_2} x_{ij_1} z_{ij_1} z_{ij_2} x_{ij_2}^\mathsf{T}.$$

For every $i = 1, \ldots, n$ and $j = 1, \ldots, m_i$, define $b_{ij} = z_{ij} x_{ij}$ and $h_{ij} = x_{ij} x_{ij}^\mathsf{T} \beta_i$. Now we can rewrite $E, F, G$ as $E = E_1 - E_2, F = F_1 - F_2, G = G_1 - G_2$, where

$$E_1 = \sum_{i=1}^{n} \frac{w_i}{m_i(m_i-1)} \left( \sum_{j=1}^{m_i} h_{ij} \right) \left( \sum_{j=1}^{m_i} h_{ij} \right)^\mathsf{T}, \qquad E_2 = \sum_{i=1}^{n} \frac{w_i}{m_i(m_i-1)} \sum_{j=1}^{m_i} h_{ij} h_{ij}^\mathsf{T},$$

$$F_1 = \sum_{i=1}^{n} \frac{w_i}{m_i(m_i-1)} \left( \sum_{j=1}^{m_i} h_{ij} \right) \left( \sum_{j=1}^{m_i} b_{ij} \right)^\mathsf{T}, \qquad F_2 = \sum_{i=1}^{n} \frac{w_i}{m_i(m_i-1)} \sum_{j=1}^{m_i} h_{ij} b_{ij}^\mathsf{T},$$

$$G_1 = \sum_{i=1}^{n} \frac{w_i}{m_i(m_i-1)} \left( \sum_{j=1}^{m_i} b_{ij} \right) \left( \sum_{j=1}^{m_i} b_{ij} \right)^\mathsf{T}, \qquad G_2 = \sum_{i=1}^{n} \frac{w_i}{m_i(m_i-1)} \sum_{j=1}^{m_i} b_{ij} b_{ij}^\mathsf{T}.$$

Define

$$r_i = \frac{(d + \log(m_i n/\delta)) \log(m_i n/\delta)}{m_i^2}, \quad s_i = \sqrt{\frac{d + \log(m_i n/\delta)}{m_i^4}} + r_i.$$

The following lemma controls the deviation of $E_1, E_2, F_1, F_2, G_1, G_2$ from their expectations $\bar{E}_1, \bar{E}_2, \bar{F}_1, \bar{F}_2, \bar{G}_1, \bar{G}_2$ in spectral norm:

**Lemma L.3.** *There exists an absolute constant $C > 0$ such that for any $\delta \in (0, 1/2)$, each of the following inequality holds with probability at least $1 - \delta$:*

$$\|E_1 - \bar{E}_1\| \leq C \sqrt{\sum_{i=1}^{n} \frac{w_i^2 \|\beta_i\|_2^4 d \log(d/\delta)}{m_i}} + C \max_i w_i \|\beta_i\|_2^2 q_i \log(d/\delta), \tag{55}$$

$$\|E_2 - \bar{E}_2\| \leq C \sqrt{\sum_{i=1}^{n} \frac{w_i^2 \|\beta_i\|_2^4 d \log(d/\delta)}{m_i^3}} + C \max_i w_i \|\beta_i\|_2^2 s_i \log(d/\delta), \tag{56}$$

$$\|F_1 - \bar{F}_1\| \leq C \sqrt{\sum_{i=1}^{n} \frac{w_i^2 \eta_i^2 \|\beta_i\|_2^2 d \log(d/\delta)}{m_i}} + C \max_i w_i \eta_i \|\beta_i\|_2 q_i \log(d/\delta), \tag{57}$$

$$\|F_2 - \bar{F}_2\| \leq C \sqrt{\sum_{i=1}^{n} \frac{w_i^2 \eta_i^2 \|\beta_i\|_2^2 d \log(d/\delta)}{m_i^3}} + C \max_i w_i \eta_i \|\beta_i\|_2 s_i \log(d/\delta), \tag{58}$$

$$\|G_1 - \bar{G}_1\| \leq C \sqrt{\sum_{i=1}^{n} \frac{w_i^2 \eta_i^4 d \log(d/\delta)}{m_i^2}} + C \max_i w_i \eta_i^2 p_i \log(d/\delta), \tag{59}$$

$$\|G_2 - \bar{G}_2\| \leq C \sqrt{\sum_{i=1}^{n} \frac{w_i^2 \eta_i^4 d \log(d/\delta)}{m_i^3}} + C \max_i w_i \eta_i^2 r_i \log(d/\delta). \tag{60}$$

Before we prove Lemma L.3, we first use it to prove Theorem L.1:

*Proof of Theorem L.1.* By (54),

$$\|A - \bar{A}\| \leq \|E_1 - \bar{E}_1\| + \|E_2 - \bar{E}_2\| + \|F_1 - \bar{F}_1\| + \|F_2 - \bar{F}_2\| + \|G_1 - \bar{G}_1\| + \|G_2 - \bar{G}_2\|.$$

Setting $\delta$ in Lemma L.3 to be $\delta/6$ and using the union bound, for an absolute constant $C > 0$, with probability at least $1 - \delta$,

$$\|A - \bar{A}\| \leq C \sqrt{\sum_{i=1}^{n} w_i^2 d \log(d/\delta) \left( \frac{\|\beta_i\|_2^4 + \eta_i^2 \|\beta_i\|_2^2}{m_i} + \frac{\eta_i^4}{m_i^2} \right)}$$

$$+ C \max_i w_i \log(d/\delta)(\|\beta_i\|_2^2 q_i + \eta_i \|\beta_i\|_2 q_i + \eta_i^2 p_i),$$

where we used the fact that the right-hand-sides of (56), (58) and (60) are upper bounded by a constant times the right-hand-sides of (55), (57), (59), respectively. Since $\bar{A} = \sum_{i=1}^n w_i \beta_i \beta_i^\mathsf{T}$, the theorem is proved by

$$\sin\theta \leq \frac{2\|A - \bar{A}\|}{\sigma_k^2}$$

due to Theorem 2.1. □

In the lemmas below, we prove helper inequalities that we use to prove Lemma L.3. In these lemmas, we focus on a single user $i$ and thus omit the subscript $i$.

**Lemma L.4.** *Let $x_1, \ldots, x_m \in \mathbb{R}^d$ are independent random vectors. Assume every $x_i$ has zero mean and is $O(1)$-sub-Gaussian. Conditioned on $x_1, \ldots, x_m$, let $z_1, \ldots, z_m \in \mathbb{R}$ be $\eta$-sub-Gaussian independent random variables with zero mean. Define $b = \sum_{j=1}^m z_j x_j$. For any $\delta \in (0, 1/2)$, with probability at least $1 - \delta$,*

$$\|b\|_2 \leq \eta \sqrt{(d + \log(1/\delta))(m + \log(1/\delta))}. \tag{61}$$

*Moreover,*

$$\|\mathbb{E}[bb^\mathsf{T}]\| \leq O(\eta^2 m), \tag{62}$$

$$\|\mathbb{E}[(bb^\mathsf{T})^2]\| \leq O(\eta^4 m^2 d). \tag{63}$$

*Proof.* Define $X = [x_1 \ \cdots \ x_m] \in \mathbb{R}^{d \times m}$ and $z = [z_1 \ \cdots \ z_m]^\mathsf{T} \in \mathbb{R}^m$. Now $b = Xz$. By Lemma C.3, with probability at least $1 - \delta/2$,

$$\|X\| \leq O\left(\sqrt{d + m + \log(1/\delta)}\right). \tag{64}$$

Let $r$ denote the rank of $X$. By Lemma C.4, with probability at least $1 - \delta/2$,

$$\|b\|_2 = \|Xz\|_2 \leq O\left(\eta\|X\|\sqrt{r + \log(1/\delta)}\right) \leq O\left(\eta\|X\|\sqrt{\min\{d, m\} + \log(1/\delta)}\right). \tag{65}$$

Combining (64) and (65) using the union bound, with probability at least $1 - \delta$,

$$\|b\|_2 \leq O\left(\eta\sqrt{(d + m + \log(1/\delta))(\min\{d, m\} + \log(1/\delta))}\right).$$

This proves (61).

For every fixed unit vector $u \in \mathbb{R}^d$,

$$u^\mathsf{T} \mathbb{E}[bb^\mathsf{T}]u = \sum_{j=1}^m u^\mathsf{T} x_j z_j^2 x_j^\mathsf{T} u = \sum_{j=1}^m z_j^2 (x_j^\mathsf{T} u)^2 \leq \sum_{j=1}^m O(\eta^2) = O(\eta^2 m).$$

This proves (62).

$$u^\mathsf{T} \mathbb{E}[(bb^\mathsf{T})^2]u = \sum_{j_1, j_2, j_3, j_4} \mathbb{E}[u^\mathsf{T} z_{j_1} x_{j_1} x_{j_2}^\mathsf{T} z_{j_2} z_{j_3} x_{j_3} x_{j_4}^\mathsf{T} z_{j_4} u]$$

Since $z_1, \ldots, z_m$ are independent and have zero mean when conditioned on $x_1, \ldots, x_m$, the term $\mathbb{E}[u^\mathsf{T} z_{j_1} x_{j_1} x_{j_2}^\mathsf{T} z_{j_2} z_{j_3} x_{j_3} x_{j_4}^\mathsf{T} z_{j_4} u]$ is zero unless $(j_1, j_2, j_3, j_4)$ belongs to

$$S := \{(j_1, j_2, j_3, j_4) \in \{1, \ldots, m\}^4 : (j_1 = j_2 \wedge j_3 = j_4) \vee (j_1 = j_3 \wedge j_2 = j_4) \vee (j_1 = j_4 \wedge j_2 = j_3)\}.$$

Letting $e_1, \ldots, e_d$ be an orthonormal bases of $\mathbb{R}^d$, we have

$$u^\mathsf{T} \mathbb{E}[(bb^\mathsf{T})^2]u = \sum_{(j_1, j_2, j_3, j_4) \in S} \sum_{\ell=1}^d \mathbb{E}[z_{j_1} z_{j_2} z_{j_3} z_{j_4} (u^\mathsf{T} x_{j_1})(x_{j_2}^\mathsf{T} e_\ell)(e_\ell^\mathsf{T} x_{j_3})(x_{j_4}^\mathsf{T} u)]$$

$$
\begin{aligned}
\leq\ & \sum_{(j_1,j_2,j_3,j_4)\in S}\sum_{\ell=1}^{d}\frac{\eta^4}{8}\mathbb{E}[(z_{j_1}/\eta)^8+(z_{j_2}/\eta)^8+(z_{j_3}/\eta)^8+(z_{j_4}/\eta)^8 \\
& + (u^{\mathsf{T}}x_{j_1})^8+(x_{j_2}^{\mathsf{T}}e_\ell)^8+(e_\ell^{\mathsf{T}}x_{j_3})^8+(x_{j_4}^{\mathsf{T}}u)^8] \\
\leq\ & \sum_{(j_1,j_2,j_3,j_4)\in S}\sum_{\ell=1}^{d}O(\eta^4) \\
\leq\ & O(\eta^4 m^2 d).
\end{aligned}
$$

This proves (63). $\qquad\square$

**Lemma L.5.** *In the same setting as the lemma above, define $h=\sum_{j=1}^{m}x_j x_j^{\mathsf{T}}\beta$ for a vector $\beta\in\mathbb{R}^d$. For any $\delta\in(0,1/2)$, with probability at least $1-\delta$,*

$$
\|h\|_2\leq O\left(\|\beta\|_2\sqrt{(d+m+\log(1/\delta))(m+\log(1/\delta))}\right). \tag{66}
$$

*Also, with probability at least $1-\delta$,*

$$
\|hh^{\mathsf{T}}-\mathbb{E}[hh^{\mathsf{T}}]\|\leq O\left(\|\beta\|_2^2\left(m\sqrt{m(d+\log(1/\delta))}+(d+\log(1/\delta))(m+\log(1/\delta))\right)\right) \tag{67}
$$

*Moreover,*

$$
\|\mathbb{E}[(hh^{\mathsf{T}}-\mathbb{E}[hh^{\mathsf{T}}])^2]\|\leq O(\|\beta\|_2^4 m^3 d). \tag{68}
$$

*Proof.* Define $X=[x_1\ \cdots\ x_m]=\mathbb{R}^{d\times m}$. We have $h=XX^{\mathsf{T}}\beta$. Since each of the $m$ entries in $X^{\mathsf{T}}\beta$ has sub-Gaussian constant $O(\|\beta\|_2)$, by Lemma C.3, with probability at least $1-\delta/2$, $\|X^{\mathsf{T}}\beta\|_2\leq O(\|\beta\|_2\sqrt{m+\log(1/\delta)})$. By Lemma C.3, with probability at least $1-\delta/2$, $\|X\|\leq O(\sqrt{d+m+\log(1/\delta)})$. This proves (66) by a union bound.

We show that with probability at least $1-\delta$,

$$
\|h-\mathbb{E}[h]\|_2\leq O\left(\|\beta\|_2\sqrt{(d+\log(1/\delta))(m+\log(1/\delta))}\right). \tag{69}
$$

If $d\geq m$, this follows from (66) and

$$
\|\mathbb{E}[h]\|_2=O(m\|\beta\|_2). \tag{70}
$$

If $d\leq m$, with probability at least $1-\delta$,

$$
\begin{aligned}
\|h-\mathbb{E}[h]\|_2 &= \|(XX^{\mathsf{T}}-\mathbb{E}[XX^{\mathsf{T}}])\beta\|_2 \\
&\leq \|XX^{\mathsf{T}}-\mathbb{E}[XX^{\mathsf{T}}]\|\cdot\|\beta\|_2 \\
&\leq O\left(\|\beta\|_2\sqrt{(d+\log(1/\delta))(m+d+\log(1/\delta))}\right) \qquad\text{(by Lemma C.2)} \\
&\leq O\left(\|\beta\|_2\sqrt{(d+\log(1/\delta))(m+\log(1/\delta))}\right).
\end{aligned}
$$

Therefore, inequality (69) holds with probability at least $1-\delta$. Now we show that $\|\mathbb{E}[h]\mathbb{E}[h]^{\mathsf{T}}-\mathbb{E}[hh^{\mathsf{T}}]\|\leq O(m\|\beta\|_2^2)$. Indeed,

$$
\mathbb{E}[h]\mathbb{E}[h]^{\mathsf{T}}-\mathbb{E}[hh^{\mathsf{T}}]=\sum_{j_1,j_2}\left(\mathbb{E}[x_{j_1}x_{j_1}^{\mathsf{T}}\beta]\mathbb{E}[x_{j_2}x_{j_2}^{\mathsf{T}}\beta]^{\mathsf{T}}-\mathbb{E}[(x_{j_1}x_{j_1}^{\mathsf{T}}\beta)(x_{j_2}x_{j_2}^{\mathsf{T}}\beta)^{\mathsf{T}}]\right).
$$

Since $x_{j_1}$ and $x_{j_2}$ are independent when $j_1\neq j_2$,

$$
\mathbb{E}[h]\mathbb{E}[h]^{\mathsf{T}}-\mathbb{E}[hh^{\mathsf{T}}]=\sum_{j=1}^{m}\left(\mathbb{E}[x_j x_j^{\mathsf{T}}\beta]\mathbb{E}[x_j x_j^{\mathsf{T}}\beta]^{\mathsf{T}}-\mathbb{E}[(x_j x_j^{\mathsf{T}}\beta)(x_j x_j^{\mathsf{T}}\beta)^{\mathsf{T}}]\right). \tag{71}
$$

It is clear that $\mathbb{E}[x_j x_j^{\mathsf{T}}\beta]=\beta$, and for every unit vector $u\in\mathbb{R}^d$,

$$
\begin{aligned}
&u^{\mathsf{T}}\mathbb{E}[(x_j x_j^{\mathsf{T}}\beta)(x_j x_j^{\mathsf{T}}\beta)^{\mathsf{T}}]u \\
&= \mathbb{E}[(u^{\mathsf{T}}x_j)(x_j^{\mathsf{T}}\beta)(\beta^{\mathsf{T}}x_j)(x_j^{\mathsf{T}}u)]
\end{aligned}
$$

$$\leq \frac{\|\beta\|_2^2}{4}\mathbb{E}[(u^\mathsf{T}x_j)^4 + (x_j^\mathsf{T}(\beta/\|\beta\|_2))^4 + ((\beta/\|\beta\|_2)^\mathsf{T}x_j)^4 + (x_j^\mathsf{T}u)^4]$$

$$\leq O(\|\beta\|_2^2).$$

This implies that $\|\mathbb{E}[(x_jx_j^\mathsf{T}\beta)(x_jx_j^\mathsf{T}\beta)^\mathsf{T}]\| \leq O(\|\beta\|_2^2)$. Now it is clear from (71) that $\|\mathbb{E}[h]\mathbb{E}[h]^\mathsf{T} - \mathbb{E}[hh^\mathsf{T}]\| \leq O(m\|\beta\|_2^2)$, which implies

$$\|hh^\mathsf{T} - \mathbb{E}[hh^\mathsf{T}]\| \leq O(m\|\beta\|_2^2) + \|hh^\mathsf{T} - \mathbb{E}[h]\mathbb{E}[h]^\mathsf{T}\|$$
$$\leq O(m\|\beta\|_2^2) + \|(h - \mathbb{E}[h])h^\mathsf{T}\| + \|\mathbb{E}[h](h - \mathbb{E}[h])^\mathsf{T}\|$$
$$\leq O(m\|\beta\|_2^2) + \|h - \mathbb{E}[h]\|_2(\|h\|_2 + \|\mathbb{E}[h]\|_2).$$

Plugging (69), (66), and (70) into the inequality above, we get (67).

Finally,
$$\mathbb{E}[(hh^\mathsf{T} - \mathbb{E}[hh^\mathsf{T}])^2] = \mathbb{E}[(hh^\mathsf{T})^2] - \mathbb{E}[hh^\mathsf{T}]^2 = \sum_{j_1,j_2,j_3,j_4} H_{j_1j_2j_3j_4}, \tag{72}$$

where

$$H_{j_1j_2j_3j_4} = \mathbb{E}[x_{j_1}x_{j_1}^\mathsf{T}\beta\beta^\mathsf{T}x_{j_2}x_{j_2}^\mathsf{T}x_{j_3}x_{j_3}^\mathsf{T}\beta\beta^\mathsf{T}x_{j_4}x_{j_4}^\mathsf{T}] - \mathbb{E}[x_{j_1}x_{j_1}^\mathsf{T}\beta\beta^\mathsf{T}x_{j_2}x_{j_2}^\mathsf{T}]\mathbb{E}[x_{j_3}x_{j_3}^\mathsf{T}\beta\beta^\mathsf{T}x_{j_4}x_{j_4}^\mathsf{T}].$$

When $j_1, j_2, j_3, j_4$ are distinct, $H_{j_1j_2j_3j_4}$ is zero. When $j_1, j_2, j_3, j_4$ are not distinct, letting $e_1, \ldots, e_d$ be an orthonormal basis for $\mathbb{R}^d$, for every fixed unit vector $u \in \mathbb{R}^d$, we have

$$u^\mathsf{T}\mathbb{E}[x_{j_1}x_{j_1}^\mathsf{T}\beta\beta^\mathsf{T}x_{j_2}x_{j_2}^\mathsf{T}x_{j_3}x_{j_3}^\mathsf{T}\beta\beta^\mathsf{T}x_{j_4}x_{j_4}^\mathsf{T}]u$$

$$= \sum_{\ell=1}^d \mathbb{E}[(u^\mathsf{T}x_{j_1})(x_{j_1}^\mathsf{T}\beta)(\beta^\mathsf{T}x_{j_2})(x_{j_2}^\mathsf{T}e_\ell)(e_\ell^\mathsf{T}x_{j_3})(x_{j_3}^\mathsf{T}\beta)(\beta^\mathsf{T}x_{j_4})(x_{j_4}^\mathsf{T}u)]$$

$$\leq \sum_{\ell=1}^d \frac{\|\beta\|_2^4}{8}\mathbb{E}[(u^\mathsf{T}x_{j_1})^8 + (x_{j_1}^\mathsf{T}(\beta/\|\beta\|_2))^8 + ((\beta/\|\beta\|_2)^\mathsf{T}x_{j_2})^8 + (x_{j_2}^\mathsf{T}e_\ell)^8$$
$$+ (e_\ell^\mathsf{T}x_{j_3})^8 + (x_{j_3}^\mathsf{T}(\beta/\|\beta\|_2))^8 + ((\beta/\|\beta\|_2)^\mathsf{T}x_{j_4})^8 + (x_{j_4}^\mathsf{T}u)^8]$$

$$= O(\|\beta\|_2^4 d).$$

This implies $\|\mathbb{E}[x_{j_1}x_{j_1}^\mathsf{T}\beta\beta^\mathsf{T}x_{j_2}x_{j_2}^\mathsf{T}x_{j_3}x_{j_3}^\mathsf{T}\beta\beta^\mathsf{T}x_{j_4}x_{j_4}^\mathsf{T}]\| = O(\|\beta\|_2^4 d)$. Similarly, $\|\mathbb{E}[x_{j_1}x_{j_1}^\mathsf{T}\beta\beta^\mathsf{T}x_{j_2}x_{j_2}^\mathsf{T}]\| = O(\|\beta\|_2^2)$ because

$$u^\mathsf{T}\mathbb{E}[x_{j_1}x_{j_1}^\mathsf{T}\beta\beta^\mathsf{T}x_{j_2}x_{j_2}^\mathsf{T}]u = \mathbb{E}[(u^\mathsf{T}x_{j_1})(x_{j_1}^\mathsf{T}\beta)(\beta^\mathsf{T}x_{j_2})(x_{j_2}^\mathsf{T}u)]$$

$$\leq \frac{\|\beta\|_2^2}{4}\mathbb{E}[(u^\mathsf{T}x_{j_1})^4 + (x_{j_1}^\mathsf{T}(\beta/\|\beta\|_2))^4 + ((\beta/\|\beta\|_2)^\mathsf{T}x_{j_2})^4 + (x_{j_2}^\mathsf{T}u)^4]$$

$$\leq O(\|\beta\|_2^2).$$

Therefore, $\|H_{j_1j_2j_3j_4}\| \leq O(\|\beta\|_2^4 d)$ when $j_1, j_2, j_3, j_4$ are not distinct. Now (68) follows from (72). $\qquad\square$

**Lemma L.6.** *In the lemma above,*
$$\|\mathbb{E}[hb^\mathsf{T}bh^\mathsf{T}]\| \leq O(\eta^2\|\beta\|_2^2 m^3 d), \text{ and } \|\mathbb{E}[bh^\mathsf{T}hb^\mathsf{T}]\| \leq O(\eta^2\|\beta\|_2^2 m^3 d). \tag{73}$$

*Proof.* For every unit vector $u$,
$$u^\mathsf{T}\mathbb{E}[hb^\mathsf{T}bh^\mathsf{T}]u = \sum_{j_1,j_2,j_3,j_4} \mathbb{E}[u^\mathsf{T}x_{j_1}x_{j_1}^\mathsf{T}\beta z_{j_2}x_{j_2}^\mathsf{T}x_{j_3}z_{j_3}\beta x_{j_4}x_{j_4}^\mathsf{T}u] \tag{74}$$

If $j_2 \neq j_3$, we have $\mathbb{E}[z_{j_2}z_{j_3}|x_{j_1}, x_{j_2}, x_{j_3}, x_{j_4}] = 0$ and thus $\mathbb{E}[u^\mathsf{T}x_{j_1}x_{j_1}^\mathsf{T}\beta z_{j_2}x_{j_2}^\mathsf{T}x_{j_3}z_{j_3}\beta x_{j_4}x_{j_4}^\mathsf{T}u] = 0$. If $j_2 = j_3$, letting $e_1, \ldots, e_d$ be an orthonormal basis for $\mathbb{R}^d$,

$$\mathbb{E}[u^\mathsf{T}x_{j_1}x_{j_1}^\mathsf{T}\beta z_{j_2}x_{j_2}^\mathsf{T}x_{j_3}z_{j_3}\beta x_{j_4}x_{j_4}^\mathsf{T}u]$$

$$= \sum_{\ell=1}^d \mathbb{E}[z_{j_2}^2(u^\mathsf{T}x_{j_1})(x_{j_1}^\mathsf{T}\beta)(x_{j_2}^\mathsf{T}e_\ell)(e_\ell^\mathsf{T}x_{j_3})(\beta^\mathsf{T}x_{j_4})(x_{j_4}^\mathsf{T}u)]$$

$$\leq \frac{\|\beta\|_2^2}{6} \sum_{\ell=1}^d \mathbb{E}[z_{j_2}^2((u^\mathsf{T}x_{j_1})^6 + (x_{j_1}^\mathsf{T}(\beta/\|\beta\|_2))^6 + (x_{j_2}^\mathsf{T}e_\ell)^6$$
$$+ (e_\ell^\mathsf{T}x_{j_3})^6 + ((\beta/\|\beta\|_2)^\mathsf{T}x_{j_4})^6 + (x_{j_4}^\mathsf{T}u)^6)]$$
$$\leq O(d\eta^2\|\beta\|_2^2).$$

Plugging this into (74) and noting that the size of $\{(j_1, j_2, j_3, j_4) \in \{1, \ldots, m\}^4 : j_2 = j_3\}$ is $O(m^3)$, we get $\|\mathbb{E}[hb^\mathsf{T}bh^\mathsf{T}]\| \leq O(\eta^2\|\beta\|_2^2 m^3 d)$.

Similarly,
$$u^\mathsf{T}\mathbb{E}[bh^\mathsf{T}hb^\mathsf{T}]u = \sum_{j_1, j_2, j_3, j_4} \mathbb{E}[u^\mathsf{T}z_{j_1}x_{j_1}\beta^\mathsf{T}x_{j_2}x_{j_2}^\mathsf{T}x_{j_3}x_{j_3}^\mathsf{T}\beta z_{j_4}x_{j_4}^\mathsf{T}u]. \tag{75}$$

If $j_1 \neq j_4$, then $\mathbb{E}[u^\mathsf{T}z_{j_1}x_{j_1}\beta^\mathsf{T}x_{j_2}x_{j_2}^\mathsf{T}x_{j_3}x_{j_3}^\mathsf{T}\beta z_{j_4}x_{j_4}^\mathsf{T}u] = 0$. When $j_1 = j_4$,

$$\mathbb{E}[u^\mathsf{T}z_{j_1}x_{j_1}\beta^\mathsf{T}x_{j_2}x_{j_2}^\mathsf{T}x_{j_3}x_{j_3}^\mathsf{T}\beta z_{j_4}x_{j_4}^\mathsf{T}u]$$
$$= \sum_{\ell=1}^d \mathbb{E}[z_{j_1}^2(u^\mathsf{T}x_{j_1})(\beta^\mathsf{T}x_{j_2})(x_{j_2}^\mathsf{T}e_\ell)(e_\ell^\mathsf{T}x_{j_3})(x_{j_3}^\mathsf{T}\beta)(x_{j_4}^\mathsf{T}u)]$$
$$\leq \frac{\|\beta\|_2^2}{6} \sum_{\ell=1}^d \mathbb{E}[z_{j_1}^2((u^\mathsf{T}x_{j_1})^6 + ((\beta/\|\beta\|_2)^\mathsf{T}x_{j_2})^6 + (x_{j_2}^\mathsf{T}e_\ell)^6$$
$$+ (e_\ell^\mathsf{T}x_{j_3})^6 + (x_{j_3}^\mathsf{T}(\beta/\|\beta\|_2))^6 + (x_{j_4}^\mathsf{T}u)^6)]$$
$$= O(d\eta^2\|\beta\|_2^2).$$

Plugging this into (75) proves $\|\mathbb{E}[bh^\mathsf{T}hb^\mathsf{T}]\| \leq O(\eta^2\|\beta\|_2^2 m^3 d)$. $\qquad\square$

We can now finish the proof of Lemma L.3.

*Proof of Lemma L.3.* To bound $\|E_1 - \bar{E}_1\|$, we set $Z_i' = \frac{w_i}{m_i(m_i-1)}\left(\sum_{j=1}^{m_i} h_{ij}\right)\left(\sum_{j=1}^{m_i} h_{ij}\right)^\mathsf{T}$ and $Z_i = Z_i' - \mathbb{E}[Z_i']$. By (67), with probability at least $1 - \delta/(2n)$,
$$\|Z_i\| \leq O\left(w_i\|\beta_i\|_2^2 q_i\right).$$

Also, by (68),
$$\|\mathbb{E}[Z_iZ_i^\mathsf{T}]\| = \|\mathbb{E}[Z_i^\mathsf{T}Z_i]\| = O\left(\frac{w_i^2\|\beta_i\|_2^4 d}{m_i}\right).$$

Lemma C.5 proves that (55) holds with probability at least $1 - \delta$.

To bound $\|E_2 - \bar{E}_2\|$, we set $Z_{ij}' = \frac{w_i}{m_i(m_i-1)}h_{ij}h_{ij}^\mathsf{T}$ and $Z_{ij} = Z_{ij}' - \mathbb{E}[Z_{ij}']$. By (67), with probability at least $1 - \delta/(2nm_i)$,
$$\|Z_{ij}\| \leq O(w_i\|\beta_i\|_2^2 s_i).$$

Also, by (68),
$$\|\mathbb{E}[Z_{ij}Z_{ij}^\mathsf{T}]\| = \|\mathbb{E}[Z_{ij}^\mathsf{T}Z_{ij}]\| = O\left(\frac{w_i^2\|\beta_i\|_2^4 d}{m_i^4}\right).$$

Lemma C.5 proves that (56) holds with probability at least $1 - \delta$.

To bound $\|F_1 - \bar{F}_1\|$, we set $Z_i = \frac{w_i}{m_i(m_i-1)}\left(\sum_{j=1}^{m_i} h_{ij}\right)\left(\sum_{j=1}^{m_i} b_{ij}\right)^\mathsf{T}$. Note that $\mathbb{E}[Z_i] = 0$. By (61) and (66), with probability at least $1 - \delta/(2n)$,
$$\|Z_i\| \leq O\left(w_i\eta_i\|\beta_i\|_2 q_i\right).$$

Also, by (73),
$$\max\left\{\|\mathbb{E}[Z_iZ_i^\mathsf{T}]\|, \|\mathbb{E}[Z_i^\mathsf{T}Z_i]\|\right\} = O\left(\frac{w_i^2\eta_i^2\|\beta_i\|_2^2 d}{m_i}\right).$$

Lemma C.5 proves that (57) holds with probability at least $1 - \delta$.

To bound $\|F_2 - \bar{F}_2\|$, we set $Z_{ij} = \frac{w_i}{m_i(m_i-1)} h_{ij} b_{ij}^\mathsf{T}$. Note that $\mathbb{E}[Z_{ij}] = 0$. By (61) and (66), with probability at least $1 - \delta/(2nm_i)$,

$$\|Z_{ij}\| \leq O(w_i \eta_i \|\beta_i\|_2 s_i).$$

Also, by (73),

$$\max\left\{\|\mathbb{E}[Z_{ij} Z_{ij}^\mathsf{T}]\|, \|\mathbb{E}[Z_{ij}^\mathsf{T} Z_{ij}]\|\right\} = O\left(\frac{w_i^2 \eta_i^2 \|\beta_i\|_2^2 d}{m_i^4}\right).$$

Lemma C.5 proves that (58) holds with probability at least $1 - \delta$.

To bound $\|G_1 - \bar{G}_1\|$, we set $Z_i' = \frac{w_i}{m_i(m_i-1)} \left(\sum_{j=1}^{m_i} b_{ij}\right)\left(\sum_{j=1}^{m_i} b_{ij}\right)^\mathsf{T}$ and $Z_i = Z_i' - \mathbb{E}[Z_i']$. By (61) and (62), with probability at least $1 - \delta/(2n)$,

$$\|Z_i\| \leq O\left(w_i \eta_i^2 p_i\right).$$

Also, by (63),

$$\|\mathbb{E}[Z_i Z_i^\mathsf{T}]\| = \|\mathbb{E}[Z_i^\mathsf{T} Z_i]\| = \|\mathbb{E}[Z_i^2]\| \leq \|\mathbb{E}[(Z_i')^2]\| = O\left(\frac{w_i^2 \eta^4 d}{m_i^2}\right).$$

Lemma C.5 proves that (59) holds with probability at least $1 - \delta$.

To bound $\|G_2 - \bar{G}_2\|$, we set $Z_{ij}' = \frac{w_i}{m_i(m_i-1)} b_{ij} b_{ij}^\mathsf{T}$ and $Z_{ij} = Z_{ij}' - \mathbb{E}[Z_{ij}']$. By (61) and (62), with probability at least $1 - \delta/(2nm_i)$,

$$\|Z_{ij}\| \leq O(w_i \eta_i^2 r_i).$$

Also, by (63),

$$\|\mathbb{E}[Z_{ij} Z_{ij}^\mathsf{T}]\| = \|\mathbb{E}[Z_{ij}^\mathsf{T} Z_{ij}]\| = \|\mathbb{E}[Z_{ij}^2]\| \leq \|\mathbb{E}[(Z_{ij}')^2]\| = O\left(\frac{w_i^2 \eta_i^4 d}{m_i^4}\right).$$

Lemma C.5 proves that (60) holds with probability at least $1 - \delta$. $\qquad\square$

## L.2  Proof of Corollary L.2

*Proof.* We note that

$$p_i \leq O\left(\frac{d \log^2(n/\delta)}{m}\right), \quad q_i \leq p_i + O\left(\sqrt{\frac{d \log(n/\delta)}{m}}\right).$$

Therefore, by Theorem L.1, with probability at least $1 - \delta$,

$$\sin\theta \leq O\left(\log(d/\delta)\sqrt{\log(n/\delta)}\sqrt{\frac{d(\frac{1}{n}\sum_{i=1}^n \|\beta_i\|_2^4 + \frac{1}{n}\sum_{i=1}^n \eta^2\|\beta_i\|_2^2 + \eta^4/m)}{mn\sigma_k^2}}\right)$$

$$+ O\left(\log(d/\delta)\log^2(n/\delta)\frac{d(\eta^2 + \max_i \|\beta_i\|_2^2)}{mn\sigma_k^2}\right).$$

Since $\sin\theta \leq 1$ always holds, the inequality above implies

$$\sin\theta \leq O\left(\log^3(nd/\delta)\left(\sqrt{\frac{d(\frac{1}{n}\sum_{i=1}^n \|\beta_i\|_2^4 + \frac{1}{n}\sum_{i=1}^n \eta^2\|\beta_i\|_2^2 + \eta^4/m)}{mn\sigma_k^4}} + \frac{d\max_i \|\beta_i\|_2^2}{mn\sigma_k^2}\right)\right)$$

$$+ O\left(\min\left\{1, \frac{d\eta^2 \log(d/\delta)\log^2(n/\delta)}{mn\sigma_k^2}\right\}\right). \tag{76}$$

Using $\frac{1}{n}\sum_{i=1}^n \|\beta_i\|_2^2 = \text{tr}(\frac{1}{n}\sum_{i=1}^n \beta_i \beta_i^\mathsf{T}) = \text{tr}(\sum_{i=1}^n w_i \beta_i \beta_i^\mathsf{T}) \geq \sum_{\ell=1}^k \sigma_\ell^2 \geq k\sigma_k^2$, we have

$$\min\left\{1, \frac{d\eta^2 \log(d/\delta)\log^2(n/\delta)}{mn\sigma_k^2}\right\} \leq \sqrt{\frac{d\eta^2 \log(d/\delta)\log^2(n/\delta)}{mn\sigma_k^2}}$$

$$\leq \sqrt{\frac{d(\frac{1}{n}\sum_{i=1}^{n}\eta^2\|\beta_i\|_2^2)\log(d/\delta)\log^2(n/\delta)}{mn\sigma_k^4}}.$$

Plugging this into (76) proves (52). When $\|\beta_i\|_2 \leq r$ for every $i = 1, \ldots, n$, (52) implies

$$\sin\theta \leq O\left(\log^3(nd/\delta)\left(\sqrt{\frac{d(r^4 + r^2\eta^2 + \eta^4/m)}{mn\sigma_k^4}} + \frac{dr^2}{mn\sigma_k^2}\right)\right).$$

Since $\sin\theta \leq 1$, the inequality above implies

$$\sin\theta \leq O\left(\log^3(nd/\delta)\left(\sqrt{\frac{d(r^4 + r^2\eta^2 + \eta^4/m)}{mn\sigma_k^4}} + \min\left\{1, \frac{dr^2}{mn\sigma_k^2}\right\}\right)\right). \qquad (77)$$

Using $r^2 \geq \frac{1}{n}\sum_{i=1}^{n}\|\beta_i\|_2^2 = \mathrm{tr}(\frac{1}{n}\sum_{i=1}^{n}\beta_i\beta_i^{\mathsf{T}}) \geq k\sigma_k^2$,

$$\min\left\{1, \frac{dr^2}{mn\sigma_k^2}\right\} \leq \sqrt{\frac{dr^2}{mn\sigma_k^2}} \leq \sqrt{\frac{dr^4}{mn\sigma_k^4}}.$$

Plugging this into (77) proves (53). $\qquad\qquad\qquad\square$