# OpenReview forum: "Subspace Recovery from Heterogeneous Data with Non-isotropic Noise"
_NeurIPS.cc/2022/Conference — NeurIPS 2022 Accept_

### Official Review · Reviewer_baLY · 2022-07-11

**Rating:** 5
**Confidence:** 3
**Soundness:** 3 good
**Presentation:** 2 fair
**Contribution:** 2 fair

**Summary:**

Consider $m = \sum_{i=1}^n m_i$ points, where for $i=1,\ldots,n$, $m_i$ (>=2) of them have been drawn independently from m_i sub-Gaussian distributions of a shared unknown mean (\mu_i) and a shared known sub-Gaussian constant (\eta_i); see Lines 184-189. Assume that m_i's and \eta_i's are known. Further, assume that \mu_1,..,\mu_n belong to a subspace. The goal is to estimate said subspace given the points.

The authors study a spectral estimator and establish recovery guarantees in terms of the maximum principal angle between the estimated subspace and the true subspace; Theorem 3.1. They also derive special cases of this guarantee under different assumptions on \mu_i's (being equal, being drawn from a certain distribution).

A matching lower bound, in certain regimes, has also been provided.

In addition to the "PCA" scenario, subspace recovery from noisy linear measurements of the \mu_i's has also been addressed.

Standard proof technique for spectral estimators, such as the one utilized in this work, has been used; outlined on top of page 6, and for the lower bound.



**Questions:**

Could you state the intuitive discussion on top of page two using a general subspace, to avoid possible confusion due to uu^T being a special case?

Please provide a proof/reference for the statement on Line 75.

Please clarify the assumption on the knowledge of eta's; e.g., in (2). It would also be helpful to provide a discussion on the availability/estimability of such information in the federated learning setup.

Please clarify the setting in which the upper and lower bounds match.

Top of page 4: please provide a brief definition for "the subspace is incoherent".

I am not sure if the "equivalence" claim on Line 238 is correct.

Providing a reference for Lines 256-272 could be helpful to readers, to connect to the rest of the relevant literature. Similarly, please provide references for the background reviewed in Appendix A; something similar to Appendix B.



**Limitations:**

Please see the comment on the assumptions in Section 3.2.


**Strengths And Weaknesses:**

I was not able to form a big picture of the multitude of assumptions in Section 3.2; on k, C*, eta_i’s, d, w_i’s, gamma_i’s, and gamma_i^prime’s. Further discussions, elaborations, or examples (special cases), on the scenario under consideration would be helpful.


While the result seem sound, I personally had a hard time following the arguments in a linear read; a sample of milestones (for the arguments and proofs) have been mentioned in the main body but I personally was not able to get a clear picture from these samples. Example: line 286 "we reduce our goal to ...", before which it would be useful to know why Gaussians are being compared. On the other hand, I think the arguments in Lines 297-321 (an entire page) can safely be summarized within the main body and moved to the appendices. I believe the current manuscript requires a revision in flow and arrangement before publication.

---

> ### Author Response · Authors · 2022-08-02
> **Author Response to Reviewer baLY**
>
> Thank you for the detailed feedback and the many good questions!
> We will certainly provide more explanations for the assumptions in Section 3.2 in the final version (as we promised in our response to reviewer sPoR as well). The key to understanding these assumptions is to view $\gamma_i$ intuitively as measuring the amount of useful information we get from user $i$. This makes sense because $\gamma_i$ increases as the number $m_i$ of data points from user $i$ increases, and $\gamma_i$ decreases as the noise magnitude $\eta_i$ from user $i$ increases. We also provided more explanations on the necessity of Assumption 3.2 in our response to reviewer sPoR, which we will also include in our final version.
>
> We answer your specific questions below.
>
> 1) **Discussion on P2 for more general subspaces.**
> We state the discussion for more general subspaces below, but we would first like to point out that the goal of the discussion is to give intuition for why non-isotropic noise is challenging and we want to explain this difficulty in the simplest possible setting with a one-dimensional subspace. Of course, a $k$-dimensional subspace for general $k$ will only make the problem harder.
>
>     The discussion generalizes straightforwardly to $k$ dimensions by replacing the unit vector $u\in \mathbb R^d$ with a matrix $U\in \mathbb O_{d,k}\subseteq \mathbb R^{d\times k}$. Specifically, suppose every $\mu_i\in\mathbb R^d$ is drawn i.i.d. from $N(0, \sigma^2 UU^{\mathsf T})$ and every data point satisfies $x_i =\mu_i + z_i$ with $z_i\in\mathbb R^d$ drawn from $N(0, \sigma^2(I - UU^{\mathsf T}) + \alpha^2 I)$. In this case every $x_i$ distributes independently from $N(0, (\sigma^2 + \alpha^2)I)$, giving no information about the $k$-dimensional subspace spanned by $\mu_1,\ldots,\mu_n$ (which is also the column space of $U$).
>
> 2) **Proof/reference for Line 75.** Our answer to the previous question gives a proof and we will make this clear in the final version.
>
> 3) **Knowledge of $\eta_i$.** That is a great question! Our estimator achieving the bound (2) needs to know $\eta_i$ to compute the weights $w_i$, but $\eta_i$ only needs to be an *upper bound* on the noise level: if the noise distribution is $\eta_i$-sub-Gaussian, it is also $\eta_i'$-sub-Gaussian for any $\eta_i'\ge \eta_i$. If we use $\eta_i'$ when computing our estimator, it would still satisfy (2) with $\eta_i$ replaced by $\eta_i'$. Obtaining this upper bound on the noise level can be much easier than estimating the exact noise level. Also, although we state our bounds for general $\eta_i$, in practice it often makes sense to assume that $\eta_i$ is the same for every user and choose the weights $w_i$ based on the number $m_i$ of data points from user $i$.
>
> 4) **In what setting does upper bound match lower bound?** Assuming $\delta = \Theta(1)$ and $d \ge (1 + \Omega(1))k$, our upper bound in Theorem 3.4 (under Assumptions 3.1 and 3.2) matches our lower bound in Theorem 3.7.
>
> 5) **Definition of incoherence.** We did not include a definition of the incoherence assumption because our results do not rely on the assumption and there are many versions of the assumption. Roughly speaking, a subspace $\Gamma \subseteq \mathbb R^d$ is incoherent if it does not "align" with any one of the basis vectors $\mathbf e_1,\ldots,\mathbf e_d$, where $\mathbf e_i\in\mathbb R^d$ is the unit vector with $i$-th coordinate being $1$. For a more formal definition, see e.g. equation (4) in Zhang et al. [2018] (cited by our paper).
>
> 6) **Equivalence at Line 238.** By Definition 3.1, we have $\gamma'_1 = \cdots = \gamma'_k = \gamma_k$, and $\gamma'_i = \gamma_i$ for every $i > k$.
>
>     Thus $\sum_{i=1}^n\gamma_i' = \sum_{i=1}^k\gamma_i' + \sum_{i=k+1}^n\gamma_i' = k\gamma_k + \sum_{i=k+1}^n\gamma_i$.
>
>     This allows us to replace the left-hand-side of the inequality in Assumption 3.2 with $k\gamma_k + \sum_{i=k+1}^n\gamma_i$ without changing the assumption. Also, we can replace $\gamma_1'$ on the right-hand-side with $\gamma_k$ because the two quantities are equal. The equivalence then follows by subtracting $k\gamma_k$ from both sides.
>
> 7) **References on the Haar measure and principal angles.** Yes, we will include references on these topics in the final version. Haar's original paper is "Der Massbegriff in der Theorie der Kontinuierlichen Gruppen" published in 1933. The idea of principal angles originated from the paper "Essai sur la géométrie à n dimensions" by Camille Jordan published in 1875.
> A good textbook on the Haar measure is "The joys of Haar measure" by Joe Diestel and Angela Spalsbury (2014).
> The textbook "Matrix Perturbation Theory" by G. Stewart and Ji-guang Sun (1990) is a good reference on principal angles.
> We will also include other useful references that we can find.

---

> > ### Comment · Reviewer_baLY · 2022-08-09
> > **reply**
> >
> > Thank you for your detailed response. A main concern in my original review was on the (technical) presentation of the results; e.g., how the assumptions have been structured, how the proof sketches have been distilled, etc. Without a revised version I am unable to comment on this aspect. However, I will change my rating to borderline accept for the next phase.

---

### Official Review · Reviewer_u5Dp · 2022-07-12

**Rating:** 6
**Confidence:** 3
**Soundness:** 3 good
**Presentation:** 3 good
**Contribution:** 3 good

**Summary:**

This paper considers the problem of subspace recovery from heterogeneous data. Specifically, it is assumed that there exists $n$ difference distributions and the $i$-th distribution generates data according to $x_i = \mu_i + z_i$ where $z_i$ is a zero-mean noise vector. The goal is to estimate the subspace spanned by ${\mu_i}_{i=1}^n$. The authors propose an estimator and provide an upper bound for its estimation error. Moreover, a matching lower bound is established to show the optimality of the estimator, which also shows that the non-spherical noise does not make the problem harder. Moving beyond, the authors also apply the method to the setting of mixed linear regression, leading to improved performance over the existing works.

**Questions:**

See above

**Strengths And Weaknesses:**

Strength:
Overall, this paper is well-written and easy to read. The theoretical results seem to be correct and sound, and the estimation error upper bound is shown to be optimal with a matching lower bound.

Weakness:
The U-statistic type estimator in (1) is not novel and similar estimators have already appeared in the literature, e.g. "On Polynomial Time Methods for Exact Low Rank Tensor Completion, Dong Xia and Ming Yuan". So I'm not very sure about the novelty of the results and This makes me believe that there might exist other works on this topic. Could the authors complete their bibliography and check if there are similar results in the literature.

---

> ### Author Response · Authors · 2022-08-02
> **Author Response to Reviewer u5Dp**
>
> Thank you for your helpful review and for pointing us to the U-statistics literature.
> Our estimators indeed use a U-statistic from every user, but the U-statistics we use are of a very basic and special form (averaging over $j_1\ne j_2$ rather than over distinct $j_1,\ldots,j_\ell$ for larger $\ell$) and our analysis does not require any techniques specific to U-statistics (such as decoupling) to achieve an optimal bound. Also, we are mostly interested in the case where each user can have as few as two data points, in which case the U-statistic from each user becomes even simpler. Xia and Yuan [2019] also use a U-statistic, but they consider a very different problem (tensor completion) and they only use a single U-statistic over many data points whereas our estimator combines many U-statistics over users with each U-statistic being possibly over very few data points.
>
> To the best of our knowledge, we are the first to study PCA with non-isotropic noise in a Federated Learning type setting with heterogeneous users. As we discussed in our related work section, prior work on PCA under non-isotropic noise essentially assumes only one data point per user and thus relies on strong additional assumptions.
>
> We will certainly add a discussion of the U-statistics literature in our final version and include Xia and Yuan [2019] in the discussion.

---

### Official Review · Reviewer_p5Rc · 2022-07-14

**Rating:** 6
**Confidence:** 3
**Soundness:** 3 good
**Presentation:** 3 good
**Contribution:** 3 good

**Summary:**

In this paper, the authors consider the problem of the principal component analysis from heterogeneous data with non-isotropic noise. The upper bound of the estimation error is established by the specific estimator, and the lower bound is obtained using Fano’s method. The upper bound matches the lower bound up to a constant factor.


**Questions:**

1. It should be explained clearly why the optimal bound established depends on the dimension $d$, but not $k$.

2. Numerical simulations like the phase transition are expected to verify the optimality of the theoretical bound.

**Limitations:**

NA.

**Strengths And Weaknesses:**

Strengths:
The main contributions of this work are the theoretical results, i.e., establishing the lower and upper bounds that match with each other up to a constant factor.

Weakness:
1. There is no numerical verification to justify the optimality of the bound established in the theorems.

2. The parameters are in the $k$-dimensional subspace. It is unclear why the optimal bound established in the theorems depends on the dimension $d$, but not $k$. It is expected to depend on $k$ instead.

---

> ### Author Response · Authors · 2022-08-02
> **Author Response to Reviewer p5Rc**
>
> Thank you for your helpful review! To answer your question on why our bounds depend on $d$ rather than $k$, we note that the dependence on $d$ is necessary even for estimating a one-dimensional subspace ($k=1$) under spherical Gaussian noise. Specifically, with $n$ users each contributing $m$ data points with i.i.d. spherical Gaussian noise,
> our lower bound (Theorem 3.7) shows that the optimal error $\sin\theta$ is at least $\Omega\left(\min\left\\{1,\sqrt{\frac{d}{mn}}\right\\}\right)$ (ignoring the dependence on parameters other than $d,m,n$). Our upper bound in Theorem 3.4 matches this lower bound up to a constant factor even for non-spherical noise.
>
> We would also like to point out that all our bounds do *not* require having at least $d$ users. Having only $k$ users is sufficient to allow $\sigma_k > 0$, in which case our error upper bounds in Section 3.1 improve towards zero as the number of data points from each user increases. Similarly, our bounds in Section 3.2 only requires $\Omega(k + \log(1/\delta))$ users if $\gamma_i$ is the same for every user (see the paragraph after Assumption 3.2). Specifically, with $n$ users each contributing $m$ data points with a constant noise level, our bound in Theorem 3.4 (also in equation (2)) becomes $\sin\theta\le O\left(\sqrt{\frac{d}{mn}}\right)$ (again ignoring the dependence on parameters other than $d,m,n$). We only need $mn\ge \Omega(d)$ to achieve a small error (e.g. $\approx \sqrt d$ users each contributing $\approx \sqrt d$ data points).
>
> Below we record results from simple numerical simulations that demonstrate how the error depends on $d$ and $k$. We generate each user-specific mean $\mu_i$ i.i.d. from $N(0, UU^{\mathsf T})$ for $U\in\mathbb O_{d,k}\subseteq \mathbb R^{d\times k}$.
> We choose the noise distribution for every user $i$ to be $N(0, V\Sigma_i V^{\mathsf T})$ with $V\in\mathbb R^{d\times d}$ drawn uniformly from $\mathbb O_d$ and $\Sigma_i$ being a diagonal matrix with each diagonal entry drawn uniformly and independently from $\\{0,1\\}$. The noise distribution is non-spherical unless $\Sigma_i = 0$ or $I$ which happens with negligible probability.
> We start with $k = 1$ and take $n = 500$ users and generate $m=10$ data points for each user.
> In the table below, we record the estimation error $\sin\theta$ of our estimator in various dimensions $d$.
> Each column in the table is the average of $10$ independent runs.
>
> |$d$|10|20|30|40|50|100|200|500|800|1000|
> |-|-|-|-|-|-|-|-|-|-|-|
> |$\sin\theta$ |0.03|0.04|0.05|0.06|0.07|0.10|0.15|0.22|0.28|0.31|
> |$\frac{\sin\theta}{\sqrt d}$|0.009|0.010|0.010|0.010|0.010|0.010|0.010|0.010|0.010|0.010|
>
> Table 1: $k=1,n=500,m=10$.
>
> The table above shows that,
> as $d$ increases, the error $\sin\theta$ grows roughly proportional to $\sqrt d$
> as predicted by our theoretical bound $\Theta\left(\sqrt{\frac{d}{mn}}\right)$.
> If we set $n = 10$ and $m = 500$ instead, we get similar errors as in the table above (in agreement with our theoretical bound $\Theta\left(\sqrt{\frac{d}{mn}}\right)$ since $mn$ does not change), which also
> shows that our estimator can indeed perform well even when the dimension $d$ becomes much larger than the number of users $n = 10$.
>
> Now, we fix $d = 500$ and study the dependence on $k$:
>
> |$k$|1|5|10|20|50|100|200|300|350|400|450|
> |-|-|-|-|-|-|-|-|-|-|-|-|
> |$\sin\theta$ |0.08|0.09|0.09|0.10|0.12|0.15|0.22|0.39|0.61|1.00|1.00|
>
> Table 2: $d=500,n=400,m=100$.
>
> The error $\sin\theta$ increases very slowly as $k$ increases from 1 to 100 (only 1.89-fold increase in error with a 100-fold increase in $k$). The error increases rapidly and reaches 1.00 as $k$ increases from 100 to 400 (a 6.62-fold increase in error with only a 4-fold increase in $k$). This demonstrates a phase transition at $k\approx 300 = 0.75 n$, showing that our assumption of having $n = \Omega(k)$ users is important. As long as this assumption is satisfied, the error dependence on $k$ is insignificant, which aligns with our theoretical bound that does not explicitly depend on $k$.
>
> Now we study the dependence on $n$ and record results in the following table. The error stays at $1$ when $n < k$ and decreases towards $0$ once $n$ exceeds $k$.
>
> |$n$|1|10|40|50|60|80|100|200|500|1000
> |-|-|-|-|-|-|-|-|-|-|-|
> |$\sin\theta$ |1.00|1.00|1.00|1.00|0.93|0.60|0.44|0.21|0.11|0.07|
>
> Table 3: $d=500,k=50,m=100$.
>
> In all of the experiments above, our estimator has extremely similar errors compared to the naive estimator which
> uses the top-$k$ eigenspace of $\sum_{i=1}^n\hat \mu_i\hat \mu_i^{\mathsf T}$ with $\hat \mu_i$ being the empirical average of points from user $i$.
> Now we change the noise distribution to be $N(0,2(I - UU^{\mathsf T}))$ and demonstrate a scenario where our estimator performs much better:
>
> |$d$|10|20|30|40|50|100|200|
> |-|-|-|-|-|-|-|-|
> |$\sin\theta$ (ours) |0.14|0.21|0.29|0.36|0.45|0.80|0.99|
> |$\sin\theta$ (naive) |0.95|0.97|0.98|0.99|0.99|0.99|1.00|
>
> Table 4: $k=1,n=500,m=2$.

---

### Official Review · Reviewer_sPoR · 2022-07-22

**Rating:** 7
**Confidence:** 3
**Soundness:** 3 good
**Presentation:** 3 good
**Contribution:** 3 good

**Summary:**

The authors consider the problem of estimating a subspace spanned by unknown vectors when given noisy observations of the vectors . Specifically, each user $i$ contains a vector $\mu_i$, and the observations are $x_{ij} = \mu_i + z_{ij}$, and the authors want to estimate the $k$-dimensional subspace spanned by $\mu_1, \cdots, \mu_n$.

The authors propose an estimator that takes a weighted combination of the observed vectors to get a sample covariance matrix, and the final subspace is that spanned by the largest $k$ eigenvectors of the sample covariance matrix. The upper bound shows that the maximal angle between the true subspace and the estimated subspace is small under certain assumptions on the sub-Gaussianity of the $\mu$ vectors.

The authors construct a hard distribution over $\mu_i$s and show an almost-matching lower bound to show that any algorithm must have an error that is close to their prescribed upper bound.

**Questions:**

I do not have major concerns about the paper. However, a more detailed discussion about the first point in the "Weaknesses" section would be appreciated.

**Limitations:**

The authors do not really address the limitations of their work. They do not state conditions under which their assumptions will fail, and addressing that is important.

**Strengths And Weaknesses:**

Strengths:
- The analysis of the upper bound is original with good technical novelty. I find the estimator and the prescribed optimal weights quite interesting.

- The lower bound uses a ``standard'' approach of using Fano's inequality on a maximally distributed set, but the construction of the hard distribution and subsequent analysis are interesting and novel.

- The application to linear models / meta-learning is interesting although it is not the main focus of the paper.

Weaknesses:
- The authors state that having non-isotropic noise in the observations is a very challenging problem, as the noise may be very correlated with the distribution of the user's vectors -- in the example provided at the beginning of the paper, the noise is strictly orthogonal to the user. However, later on, they assume that the noises are subGaussian, and the user vectors are sufficiently anti-concentrated within their subspace, and now it's not clear whether the same hardness holds.

- The writing can be a little confusing at times. The bound in Theorem 2.1 requires $\sigma_k$ to depend on $w$, and this is only resolved in the next section -- even there, the authors do not state how $\sigma_k$ is ultimately lower bounded.

- The assumptions can be somewhat cryptic, such as Assumption 3.2. I suppose Assumption 3.2 helps ensure that users $k+1, \cdots , n$ (after sorting by relative importance) provide useful information. I'm not sure the paragraph after Assumption 3.2 helps explain it's significance / requirement.

---

> ### Author Response · Authors · 2022-08-02
> **Author Response to Reviewer sPoR**
>
> Thank you for your thoughtful feedback and the many helpful comments! We will make sure to make refinements based on these comments in the final version. Below we would like to respond to your three points in the "weaknesses" section in more detail.
>
> For your first point in the "weaknesses" section, we would like to emphasize that our sub-Gaussian assumption on the noises does not require the noises to be isotropic. In particular our example at the beginning of the paper satisfies this assumption: the non-isotropic noise distribution $N(0,\sigma^2(I - uu^{\mathsf T}) + \alpha^2I)$ is an $\eta = O(\sqrt{\sigma^2 + \alpha^2})$-sub-Gaussian distribution. When each user has $m_i = 2$ data points, our error bound (2) applies to this case and gives $\sin\theta \le O\left(\frac{\sigma^2 + \alpha^2}{\sigma^2}\sqrt{\frac{d + \log(1/\delta)}{n}}\right)$, where the error approaches zero as the number $n$ of users grows. In contrast, with only one data point per user, the error does not decrease no matter how large $n$ grows because we get no useful information from the data.
> Our sub-Gaussian assumption only controls the noise magnitude and does not restrict the shape of the noise. Clearly, some assumption on the noise magnitude is necessary, and our analysis needs the sub-Gaussian assumption which is also commonly used in other statistical estimation problems. We leave the interesting question of using an even weaker assumption to future work.
>
> We would also like to point out that we always allow the noise $z_{ij}$ to depend on the means $\mu_{i'}$ even for $i'\ne i$, although we require the conditional distribution of $z_{ij}$ given $\mu_{i'}$ to be mean-zero and sub-Gaussian. We will make this clearer in the final version.
>
> Your second and third points in the "weaknesses" section are very helpful for us and we will improve our presentation based on these comments, but we would like to point out that "$\sigma_k$ is ultimately lower bounded" in Claim 3.3 where we show $\sigma_k^2\ge \sigma^2/2$ with probability $\ge 1-\delta/2$ under our assumptions. As you pointed out, the quantity $\gamma_i$ intuitively signifies the amount of useful information we get from user $i$, and Assumption 3.2 requires that a significant fraction of that information comes from outside the $k$ most informative users. This assumption is necessary because if we only had exactly $n=k$ users, we would have $\sigma_k^2\approx\sigma^2/k^2$ (instead of the desired $\sigma_k^2\ge \sigma^2/2$) even for uniform weights $w_1 = \cdots = w_n$ (see e.g. [1]), where having $n = C_*(k + \log(1/\delta))$ users for a constant $C_* > 1$ would immediately save the loss of the $\Theta(k^2)$ factor in $\sigma_k^2$. We will include this explanation in the final version.
>
> [1] Mark Rudelson, and Roman Vershynin.  "The Littlewood–Offord problem and invertibility of random matrices." Advances in Mathematics 218.2 (2008)

---

### Meta-Review · Area_Chair_caRb · 2022-08-25

**Recommendation:** Accept
**Confidence:** Certain

**Metareview:**

This paper studies the problem of performing subspace recovery (i.e. PCA) with heterogenous and non-isotropic noise. In particular there are $n$ users who each get samples drawn from a $d$ dimensional distribution with mean $\mu_i$. Furthermore the means lie in a $k$ dimensional subspace, and the goal is to estimate it. The main catch is that while they require the noise to be subgaussian, they make no assumption on it being isotropic or homogenous across the different users. When each user gets only one sample, the problem is impossible. But when each user gets two samples, they give a simple estimator based on appropriately chosen $U$-statistics and bound its estimation error. Moreover they show that this bound is optimal up to constant factors. This is the first work to study PCA in a federated setting. It is a clean problem, with an elegant and complete solution.

**Award:**

No

---

### Decision · Program_Chairs · 2022-09-14

Accept